# Differential regulation by CD47 and thrombospondin-1 of extramedullary erythropoiesis in mouse spleen

Rajdeep Banerjee[1], Thomas J Meyer[2], Margaret C Cam[2], Sukhbir Kaur[1], David D Roberts[1]*

[1]Laboratory of Pathology, Center for Cancer Research, National Cancer Institute, National Institutes of Health, Bethesda, United States; [2]CCR Collaborative Bioinformatics Resource, Office of Science and Technology Resources, National Cancer Institute, National Institutes of Health, Bethesda, United States

**Abstract** Extramedullary erythropoiesis is not expected in healthy adult mice, but erythropoietic gene expression was elevated in lineage-depleted spleen cells from *Cd47*[−/−] mice. Expression of several genes associated with early stages of erythropoiesis was elevated in mice lacking CD47 or its signaling ligand thrombospondin-1, consistent with previous evidence that this signaling pathway inhibits expression of multipotent stem cell transcription factors in spleen. In contrast, cells expressing markers of committed erythroid progenitors were more abundant in *Cd47*[−/−] spleens but significantly depleted in *Thbs1*[−/−] spleens. Single-cell transcriptome and flow cytometry analyses indicated that loss of CD47 is associated with accumulation and increased proliferation in spleen of Ter119[−]CD34[+] progenitors and Ter119[+]CD34[−] committed erythroid progenitors with elevated mRNA expression of Kit, Ermap, and Tfrc. Induction of committed erythroid precursors is consistent with the known function of CD47 to limit the phagocytic removal of aged erythrocytes. Conversely, loss of thrombospondin-1 delays the turnover of aged red blood cells, which may account for the suppression of committed erythroid precursors in *Thbs1*[−/−] spleens relative to basal levels in wild-type mice. In addition to defining a role for CD47 to limit extramedullary erythropoiesis, these studies reveal a thrombospondin-1-dependent basal level of extramedullary erythropoiesis in adult mouse spleen.

*For correspondence: droberts@mail.nih.gov

Competing interest: The authors declare that no competing interests exist.

## eLife assessment

This study presents a **valuable** finding on the cell composition in mouse spleen depleted for the CD47 receptor and its signaling ligand Thrombospondin in hematopoietic differentiation. The supporting evidence is **convincing** with analytical improvements on the individual contributions of the signaling components and with functional studies. This work has implications for the role of CD47/Thbs1 in extramedullary erythropoiesis in mouse spleen and will be of interest to medical biologists working on cell signaling, transfusion medicine, and cell therapy.

## Introduction

CD47 is a counter-receptor for signal-regulatory protein-α (SIRPα; *Matozaki et al., 2009*) and a component of a supramolecular membrane signaling complex for thrombospondin-1 that contains specific integrins, heterotrimeric G proteins, tyrosine kinase receptors, exportin-1, and ubiquilins (*Gao et al., 1996*; *Isenberg et al., 2009*; *Kaur et al., 2022*; *Soto-Pantoja et al., 2015*). CD47 binding to SIRPα on macrophages induces inhibitory signaling mediated by its cytoplasmic immunoreceptor

tyrosine-based inhibition motifs that recruit and activate the tyrosine phosphatases SHP-1 and SHP-2 (**Matozaki et al., 2009**). Loss of this inhibitory signaling results in rapid splenic clearance of $Cd47^{-/-}$ mouse red blood cells (RBC) when transfused into a wild type (WT) recipient (**Oldenborg et al., 2000**). The species-specificity of CD47/SIRPα binding constitutes a barrier to interspecies blood transfusion and hematopoietic reconstitution (**Strowig et al., 2011**; **Wang et al., 2007**).

CD47 forms nanoclusters on young RBC with limited binding to thrombospondin-1 (**Wang et al., 2020**). CD47 abundance decreases on aged RBCs, but CD47 on aging RBC forms larger and more dense clusters with increased ability to bind thrombospondin-1. CD47 on aging RBC also adopts an altered conformation (**Burger et al., 2012**). Exposure of aged RBC to thrombospondin-1 further increases the size of CD47 clusters via a lipid-raft-dependent mechanism. Conversely, CD47 cluster formation was limited on $Thbs1^{-/-}$ mouse RBC and associated with significantly increased RBC lifespan (**Wang et al., 2020**).

Liver and spleen are the main hematopoietic organs during embryonic development, whereas bone marrow assumes that responsibility after birth (**Kim, 2010**). Induction of extramedullary hematopoiesis in adult spleen can compensate for pathological conditions that compromise hematopoiesis in bone marrow (**Cenariu et al., 2021**). CD47 is highly expressed on proliferating erythroblasts during stress-induced erythropoiesis, and antibodies blocking either CD47 or SIRPα inhibited the required transfer of mitochondria from macrophages to developing erythroblasts in erythroblastic islands (**Yang et al., 2022**). Notably, treatment with a CD47 antibody enhanced splenomegaly in the anemic stress model. Consistent with the absence of inhibitory SIRPα signaling that limits clearance of aged RBC (**Fossati-Jimack et al., 2002**; **Lutz and Bogdanova, 2013**), $Cd47^{-/-}$ mice derived using CRISPR/Cas9 exhibited hemolytic anemia and splenomegaly (**Kim et al., 2018**). Conversely, CD47-dependent thrombospondin-1 signaling regulates the differentiation of multipotent stem cells in a stage-specific manner (**Kaur et al., 2013**; **Nath et al., 2018**; **Porpiglia et al., 2022**). and both $Thbs1^{-/-}$ and $Cd47^{-/-}$ mouse spleens have more abundant Sox2⁺ stem cells and higher mRNA expression of the multipotent stem cell transcription factors Myc, Sox2, Oct4, and Klf4 (**Kaur et al., 2013**). Therefore, both thrombospondin-1- and SIRPα-dependent CD47 signaling could alter erythropoiesis and contribute to spleen enlargement. Here, we utilized flow cytometry combined with bulk and single-cell transcriptomics to examine extramedullary hematopoiesis in $Cd47^{-/-}$ and $Thbs1^{-/-}$ mice, which revealed cooperative and opposing roles for CD47 and thrombospondin-1 to limit extramedullary erythropoiesis in spleen.

## Results

### Upregulation of erythroid precursors in $Cd47^{-/-}$ mouse spleen

We confirmed the previously reported spleen enlargement in $Cd47^{-/-}$ mice (**Bian et al., 2016**; **Nath et al., 2018**), but we did not observe significant spleen enlargement in $Thbs1^{-/-}$ mice (**Figure 1—figure supplement 1A**). Enlargement of $Cd47^{-/-}$ spleens could result from increased phagocytic clearance of RBC, as reported in $Cd47^{-/-}$ and $Sirpa^{-/-}$ mice treated with CpG (**Kidder et al., 2020**) and aging $Cd47^{-/-}$ mice (**Kim et al., 2018**). Alternatively, increased cell numbers could result from the increased stem cell abundance in $Cd47^{-/-}$ spleens (**Kaur et al., 2013**). The spleen enlargement was associated with a significantly higher total spleen cell number in a single cell suspension after RBC lysis in $Cd47^{-/-}$ mice compared to WT and $Thbs1^{-/-}$ mice (**Figure 1A**).

Our previous analysis of lineage-negative cells from WT and $Cd47^{-/-}$ spleens identified an increased abundance of NK cell precursors in $Cd47^{-/-}$ spleens (**Nath et al., 2018**). Analysis of bulk RNAseq data of naïve WT and $Cd47^{-/-}$ spleen cells depleted for proerythroblasts through mature erythrocytes using the antibody Ter-119 (**Kina et al., 2000**) and for cells bearing CD4, CD11b, CD11c, CD19, CD45R, CD49b, CD105, MHC Class II, and TCRγ/δ (Lin⁻CD8⁺) unexpectedly showed strong enrichment of a heme metabolism gene signature (**Figure 1—figure supplement 1C**), markers of stress-induced erythropoiesis (**Delic et al., 2020**; **Thompson et al., 2010**) and adult definitive erythropoiesis (**Kingsley et al., 2013**) in the lineage-depleted $Cd47^{-/-}$ relative to the corresponding cells from WT spleens (**Table 1**, **Figure 1—figure supplement 1B and D**). Trim10 mRNA, which encodes an erythroid-specific RING finger protein required for terminal erythroid differentiation (**Harada et al., 1999**), was elevated 50-fold. Higher Mki67 mRNA expression suggested increased proliferation among Lin⁻ $Cd47^{-/-}$ spleen cells, which also expressed elevated mRNA levels of the major erythroid transcription factor Gata1 (**Gutiérrez et al., 2020**). Apart from increased Kit mRNA expression, however, mRNA expression of

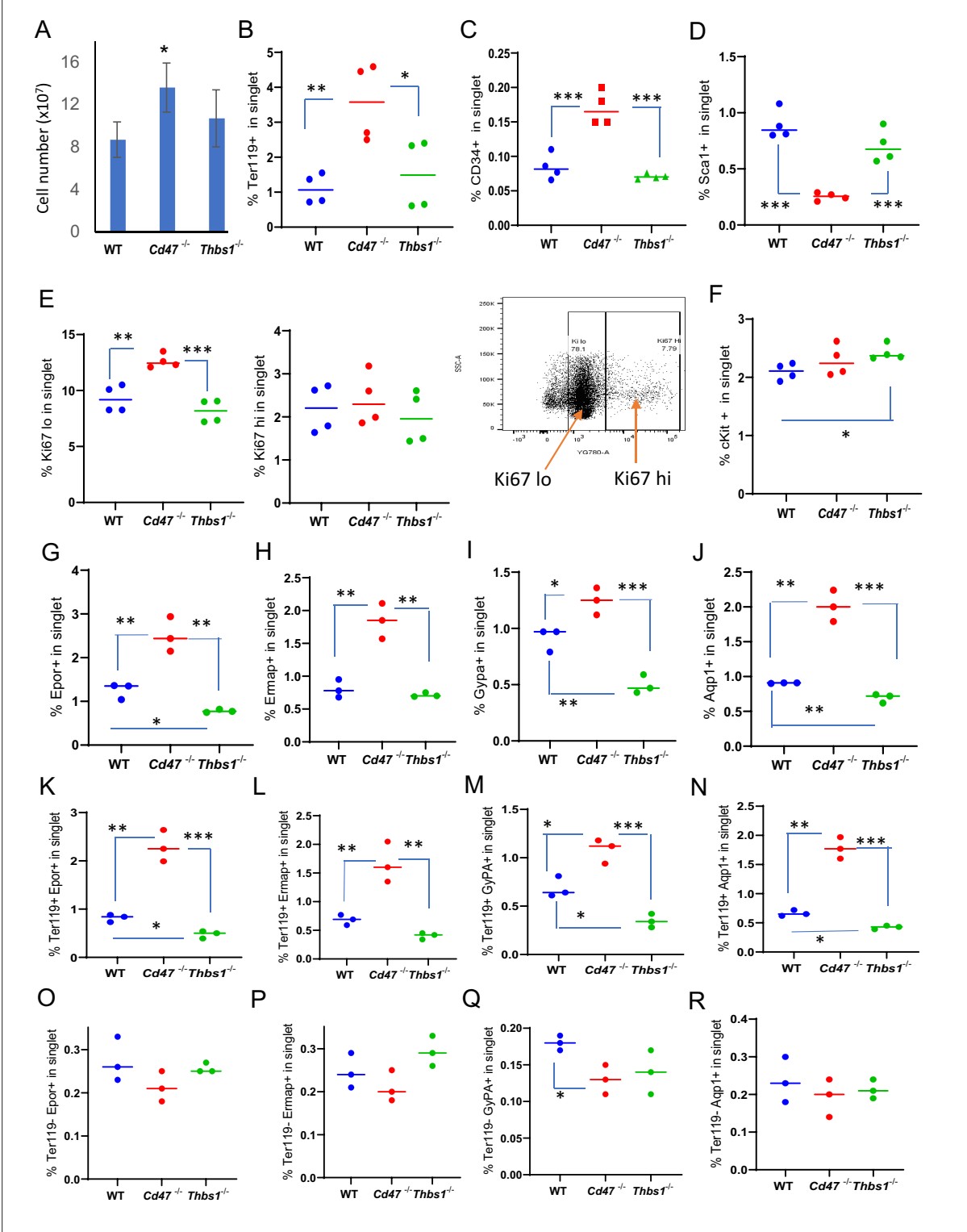

**Figure 1.** Effects of *Cd47* or *Thbs1* gene disruption on spleen cell numbers and content of cells expressing erythropoietic precursor markers or the proliferation marker Ki67. (**A**) Total spleen cell numbers in WT, *Cd47⁻/⁻* and *Thbs1⁻/⁻* C57BL/6 mice determined after lysis of RBC (mean ± SEM, n=3). Flow cytometry was performed to analyze gated singlet spleen cells stained with Ter119 antibody (**B**), CD34 antibody (**C**), Sca1 antibody (**D**), Ki67 antibody with the indicated gating for high and low expression (**E**), cKit antibody (**F**), Epor antibody (**G**), Ermap antibody (**H**), Gypa antibody (**I**), or Aqp1 antibody (**J**). Further analysis of Epor (**K, O**), Ermap (**L, P**), Gypa (**M, Q**), and Aqp1 expression (**N, R**) was performed after gating for Ter119 expression.

*Figure 1 continued on next page*

*Figure 1 continued*

The percentages of cells positive for the indicated surface markers are presented (n=3 or 4 mice of each genotype). p-values were determined using a two-tailed t test for two-samples assuming equal variances in GraphPad Prism. *=p < 0.05, **=p < 0.01, ***=p < 0.001.

The online version of this article includes the following figure supplement(s) for figure 1:

**Figure supplement 1.** Enlargement of spleen in the absence of CD47 and bulk RNAseq analysis of lin− *Cd47⁻/⁻* vs WT spleen cells.

**Figure supplement 2.** Flow cytometry analysis strategy.

markers for multipotent erythroid progenitors including Anpep (CD13), Cd33, Sca1 and Gata2 was not elevated in the absence of CD47. These data suggested preferential accumulation of committed erythroid progenitors rather than multipotent erythroid progenitors in the *Cd47⁻/⁻* spleens. CD47 regulates activities of the nuclear transport protein exportin-1 (***Kaur et al., 2022***), and Xpo1 mRNA was also increased in *Cd47⁻/⁻* cells (***Table 1***). Exportin-1 is a Gata1 transcriptional target and promotes

**Table 1.** Expression of erythropoiesis-associated genes in lineage-depleted *Cd47⁻/⁻* versus WT spleen cells.

| Gene | Erythropoiesis expression/function | Fold change *Cd47⁻/⁻/WT*[*] | T statistic | p-value |
|------|-----------------------------------|---------------------------|-------------|---------|
| *Ermap* | extramedullary erythropoiesis marker[†] | 21.5 | 6.04 | $9.35 \times 10^{-5}$ |
| *Tal1* | extramedullary erythropoiesis marker[†] | 13.9 | 5.13 | $3.55 \times 10^{-4}$ |
| *Gypa* | extramedullary erythropoiesis marker[†] | 89.3 | 3.67 | $3.86 \times 10^{-3}$ |
| *Gata1* | extramedullary erythropoiesis marker[†] | 7.69 | 4.86 | $5.38 \times 10^{-4}$ |
| *Kel* | extramedullary erythropoiesis marker[†] | 29.0 | 4.61 | $8.04 \times 10^{-4}$ |
| *Slc4a1* | extramedullary erythropoiesis marker[†] | 151.4 | 4.90 | $5.05 \times 10^{-4}$ |
| *Klf1* | extramedullary erythropoiesis marker[†] | 20.6 | 5.06 | $3.95 \times 10^{-4}$ |
| *Cldn13* | extramedullary erythropoiesis marker[†] | 49.2 | 4.42 | $1.09 \times 10^{-3}$ |
| *Trim10* | extramedullary erythropoiesis marker[†] | 49.7 | 4.11 | $1.83 \times 10^{-3}$ |
| *Epor* | extramedullary erythropoiesis marker[†] | 12.3 | 4.91 | $5.01 \times 10^{-4}$ |
| *Sptb* | extramedullary erythropoiesis marker[†] | 36.5 | 5.21 | $3.16 \times 10^{-4}$ |
| *Rhag* | extramedullary erythropoiesis marker[†] | 33.0 | 4.64 | $7.68 \times 10^{-4}$ |
| *Hba-a1* | erythroblasts | 63.8 | 6.46 | $5.23 \times 10^{-5}$ |
| *Hbb-bs* | erythroblasts | 59.2 | 6.60 | $4.34 \times 10^{-5}$ |
| *Gata1* | BFU-E through erythroblasts | 7.69 | 4.86 | $5.38 \times 10^{-4}$ |
| *Tfrc* (CD71) | CFU-E through erythroblasts | 3.12 | 6.19 | $7.58 \times 10^{-5}$ |
| *Kit* | Progenitors through CFU-E | 1.72 | 6.24 | $7.02 \times 10^{-5}$ |
| *Sox6* | Adult definitive erythropoiesis | 35.5 | 3.56 | 0.0046 |
| *Aqp1* | Adult definitive erythropoiesis | 26.9 | 6.51 | $4.91 \times 10^{-5}$ |
| *Nr3c1* | Adult definitive erythropoiesis | 1.12 | 2.80 | 0.018 |
| *Mki67* | Proliferation marker | 4.43 | 7.65 | $1.14 \times 10^{-5}$ |
| *Cd34* | Multipotent progenitors through CFU-E | 1.60 | 1.59 | 0.141 |
| *Ly6a* (Sca1) | Multipotent progenitors | 1.17 | 1.02 | 0.331 |
| *Anpep* (CD13) | Multipotent progenitors | 1.16 | 1.14 | 0.28 |
| *Cd33* | Multipotent progenitors | −1.06 | −0.37 | 0.71 |
| *Gata2* | Multipotent progenitors | 1.08 | 0.66 | 0.52 |
| *Xpo1* | Stability of nuclear Gata1 | 1.23 | 3.86 | $2.7 \times 10^{-3}$ |

[*]Gene enrichment in naïve *Cd47⁻/⁻* vs WT spleen cells depleted for CD4, CD11b, CD11c, CD19, CD45R (B220), CD49b (DX5), CD105, MHC Class II, Ter-119, and TCRγ/δ.

[†]Reported markers of stress-induced extramedullary erythropoiesis (***Delic et al., 2020***; ***Thompson et al., 2010***).

terminal erythroid differentiation by maintaining Gata1 in the nucleus (*Guillem et al., 2020*), which suggested a potential mechanism by which loss of CD47 could increase erythropoiesis.

To further characterize CD47-dependent spleen cells and the relevance of thrombospondin-1, single cell suspensions of depleted of mature RBC were analyzed using flow cytometry for expression of erythropoiesis-related cell surface and proliferation markers (*Figure 1B–J*, *Figure 1—figure supplement 2*). The percentages of Ter119+ cells in singlet cells from *Cd47−/−* spleens was significantly higher than in WT or *Thbs1−/−* spleens (p=0.0061 and 0.0322 respectively, *Figure 1B*). CD34 is expressed on multipotent through the CFU-E erythroid progenitors and was expressed a significantly higher percentage on *Cd47−/−* versus WT or *Thbs1−/−* cells (p=0.0008 and 0.0001 respectively, *Figure 1C*), whereas the multipotent progenitor marker Sca1 was expressed in a smaller percentage of *Cd47−/−* versus WT or *Thbs1−/−* spleen cells (p=0.0001 and 0.0009 respectively, *Figure 1D*).

A higher percentage of *Cd47−/−* cells expressed the proliferation marker Ki67 at low but not high levels compared to WT or *Thbs1−/−* spleen cells (p=0.0024 and 0.0003 respectively, *Figure 1E*). c-Kit signaling is crucial for normal hematopoiesis and is expressed in multipotent progenitors through CFU-E (*Lennartsson and Rönnstrand, 2012*; *Swaminathan et al., 2022*), but the percentage of cKit positive cells was higher only in *Thbs1−/−* spleen cells (*Figure 1F*).

Erythrocyte lineage markers including Ermap, glycophorin A (Gypa), Epor and Aqp1 are established markers of stress-induced extramedullary erythropoiesis (*Delic et al., 2020*). The erythropoietin receptor (Epor), which is expressed in CFU-E through proerythroblasts, was expressed in significantly more *Cd47−/−* versus WT spleen cells (p=0.0077) but in significantly fewer *Thbs1−/−* spleen cells (p=0.0017 *Figure 1G*). Cells expressing the major RBC membrane glycoprotein Gypa, which accumulates in erythroblasts, Ermap, and Aqp1 showed similar significant increases in *Cd47−/−* spleen cells, whereas only Gypa+ and Aqp1+ cells were significantly decreased in *Thbs1−/−* spleen cells (*Figure 1H, I and J*). Consistent with their known induction kinetics, most of the cells expressing these markers were Ter119+, and the alterations in their abundance observed in *Cd47−/−* and *Thbs1−/−* spleens were restricted to Ter119+ cells (*Figure 1K–R*). These results indicate an increased abundance of committed erythroid progenitors spanning CD34+ progenitors through reticulocytes in the *Cd47−/−* spleen but depletion of the earlier Sca1+ multipotent progenitors. Consistent with the decreased turnover of *Thbs1−/−* RBC (*Wang et al., 2020*), cells expressing the committed erythroid markers Gypa, Epor, and Aqp1 were depleted in *Thbs1−/−* spleens.

Erythropoietin binding to Epor on early erythroid precursor cells stimulates their survival, proliferation, and differentiation by inducing the master transcriptional regulators Tal1, Gata1, and Klf1 (*Perreault and Venters, 2018*). Tal1, Gata1, and Klf1, and Epor mRNAs were strongly up-regulated in *Cd47−/−* relative to WT spleens (*Table 1*). The increased proliferative response in the *Cd47−/−* cells is consistent with a significant increase in Epor+ Ter119+ *Cd47−/−* cells, contrasting with a significant decrease in their abundance in Ter119+ *Thbs1−/−* cells (*Figure 1K*). Epor expression was minimal and unchanged in the Ter119− population (*Figure 1O*), indicating that the Ter119 antigen is expressed in erythroid precursors that are responsive to erythropoietin. These data are consistent with increased extramedullary erythropoiesis in *Cd47−/−* and suppressed extramedullary erythropoiesis relative to WT in *Thbs1−/−* spleens.

## CD47-dependence of erythropoietic markers in Ter119+ and Ter119− cells

The Ter119 antibody recognizes an antigen highly expressed in mouse proerythroblasts through mature erythrocytes (*Kina et al., 2000*). Although Ter119 is a widely used erythroid lineage marker, its epitope does not map to a specific protein (*Kina et al., 2000*) and requires 9-O-acetylation of sialic acids that are present on several RBC glycoproteins (*Mahajan et al., 2019*). Although no effect of CD47 on the abundance of cKit+ cells was detected (*Figure 1F*), Kit mRNA was significantly enriched in lineage-negative *Cd47−/−* spleen cells (*Table 1*). To resolve this discrepancy, the percentage of cKit+ cells was assessed in Ter119+ and Ter119− populations (*Figure 2A*, *Figure 2—figure supplement 1*). A higher percentage of the *Cd47−/−* cells were Ter119+cKit+. Consistent with *Figure 1F* and the elevation of multipotent stem cells in *Thbs1−/−* spleen (*Kaur et al., 2013*), Ter119−cKit+ cells were elevated in *Thbs1−/−* spleens. Consistent with the bulk RNA sequencing data, Sca1+ cells in spleen did not differ in the Ter119+ cells and were less abundant in Ter119− cells from *Cd47−/−* spleens relative to the same subset from WT (*Figure 2B*). Consistent with the negative regulation of stem cell transcription factors

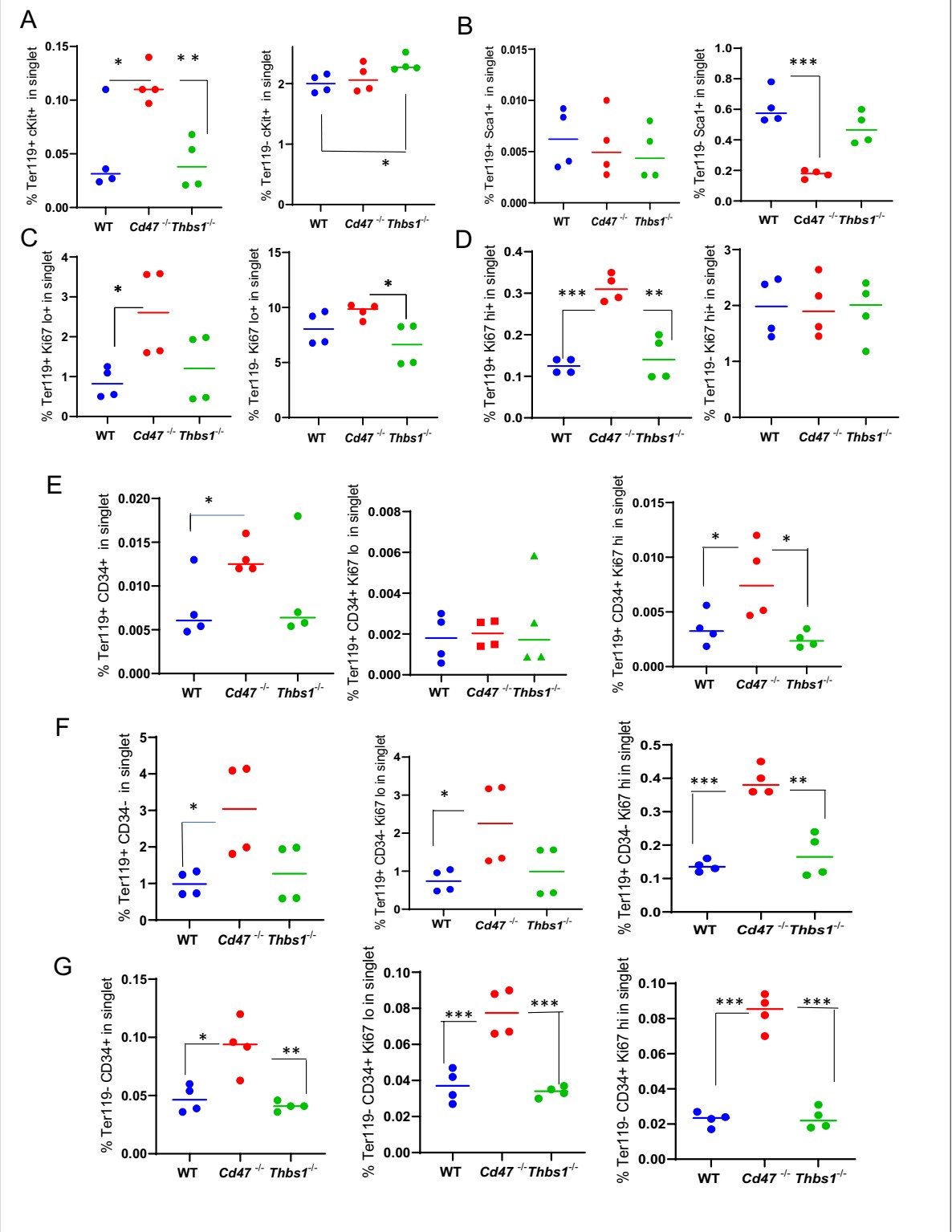

**Figure 2.** Effects of *Cd47* or *Thbs1* gene disruption on the percentage of Ter119+ and Ter119− spleen cells expressing markers of multipotent and committed erythroid precursors and cell proliferation. Spleen cells isolated from WT, *Cd47*−/− and *Thbs1*−/− mice (2 male and 2 female of each genotype) were costained with Ter119 antibody along with cKit, Ki67, Sca1 Ermap, Gypa, Epor, or Aqp1 antibodies and acquired on an LSRFortessa SORP. After gating for singlet cells, the percentages of Ter119+ and Ter119− cells positive for stem cell markers cKit (**A**) and Sca1 (**B**), high or low levels of the proliferation marker Ki67 (**C, D**), were compared among WT, *Cd47*−/− and *Thbs1*−/− mouse spleens (n=3–4). The proliferation of CD34+ and CD34− populations of Ter119+ spleen cells from WT, *Cd47*−/−, *Thbs1*−/− mice was evaluated by staining with CD34, Ter119 and Ki67 antibodies. Ter119+CD34+

*Figure 2 continued on next page*

*Figure 2 continued*

cells (**E**), Ter119⁺CD34⁻ cells (**F**), and Ter119⁻CD34⁺ cells (**G**) were also quantified (left panels) and analyzed for the proliferation marker Ki67 (center and right panels). p-values were determined using a two-tailed t test for two-samples assuming equal variances in GraphPad Prism. *=p < 0.05, **=p < 0.01, ***=p < 0.001.

The online version of this article includes the following figure supplement(s) for figure 2:

**Figure supplement 1.** Flow cytometry analysis strategy.

in spleen by CD47-dependent thrombospondin-1 signaling (*Kaur et al., 2013*), these results support prior evidence that that loss of *Thbs1* or *Cd47* results in accumulation of early hematopoietic precursors that are Ter119⁻ and demonstrates a *Cd47⁻/⁻*-specific enrichment of cKit⁺Ter119⁺ committed erythroid precursors. Proliferating cells with low Ki67 expression were more abundant in the Ter119⁺ and Ter119⁻ populations of *Cd47⁻/⁻* cells relative to WT and *Thbs1⁻/⁻* cells, suggesting that loss of *Cd47* but not *Thbs1* increases the proliferation of committed erythroid precursors (*Figure 2C and D*).

## CD47 limits proliferation of Ter119⁺CD34⁻ and Ter119⁻CD34⁺ spleen cells

Although *Figure 1* demonstrated increased numbers of Ter119⁺ and CD34⁺ cells in *Cd47⁻/⁻* spleens, further analysis revealed that more CD34⁺ cells are Ter119⁻ than Ter119⁺ (*Figure 2E and G*, left panels). Therefore, loss of CD47 upregulates both early Ter119⁻CD34⁺ progenitors and more mature Ter119⁺ progenitors, most of which have lost CD34 expression (*Figure 2F*). Most of the increased proliferation of *Cd47⁻/⁻* spleen cells, indicated by Ki67 expression, was in the Ter119⁺CD34⁻ subset (*Figure 2F*, center and right panels), but proliferating *Cd47⁻/⁻* cells were also enriched in the Ter119⁻CD34⁺ population (*Figure 2G*). These data indicate roles for more differentiated Ter119⁺CD34⁻ erythroid progenitors as well as earlier Ter119⁻CD34⁺ erythroid progenitors in mediating the increased extramedullary erythropoiesis in *Cd47⁻/⁻* spleens.

## Identification of CD47-dependent erythroid precursor populations

Single cell RNA sequencing (scRNAseq) was used to further define effects of CD47 and thrombospondin-1 on erythroid precursors in spleen (*Figure 3*, *Figure 3—figure supplement 1*). Spleen cells from WT, *Cd47⁻/⁻* and *Thbs1⁻/⁻* mice were treated with immobilized antibodies to deplete monocytic, T, B, and NK cell lineages and mature RBC and subjected to scRNAseq analysis. Following alignment, the mRNA expression data were clustered in two dimensions using the t-distributed stochastic neighbor embedding (tSNE) method in NIDAP. Using a resolution of 0.4, the spleen cells clustered in 18 groups (*Figure 3A*).

Cell type analysis using SingleR with Immgen and mouse RNAseq databases identified the main cell types in each cluster (*Figure 3B*). Immgen main cell type annotation identified cluster 12 and a subset of cluster 14 as stem cells, and mouse RNA seq annotation identified erythrocyte signatures mostly in cluster 12. WT, *Cd47⁻/⁻*, and *Thbs1⁻/⁻* cells clustered by genotype in the major residual T cell clusters 0 through 5 but had similar distributions within clusters 12 and 14 (*Figure 3C and D*).

Erythroid lineage markers were found mainly in cluster 12, which contained 440 cells (*Figure 4* and *Figure 4—figure supplements 1 and 2*). Consistent with the flow data in *Figures 1 and 2*, *Cd47⁻/⁻* cells were more abundant in cluster 12 (65%), and *Thbs1⁻/⁻* cells were less abundant (15%) than WT cells (21%). CD34 was expressed mostly in cluster 14 and to lower parts of cluster 12, whereas Ly6a (Sca1) was restricted to isolated cells in both clusters. Kit and Gata2, which are expressed by multipotent progenitors through CFU-E, were expressed in lower and middle areas of cluster 12 and to a limited degree in cluster 14 (*Figure 4*). The extramedullary erythropoiesis markers Epor, Gata1, Klf1, and Ermap, the definitive erythropoiesis marker Aqp1 and the proliferation marker Ki67 had similar distributions in cluster 12 (*Figure 4*). Trim10 and the late erythroid markers Gypa, Tmem56, Epb42, Spta1, and Sptb were restricted to the upper regions of cluster 12 (*Figure 4*). Therefore, erythroid differentiation within cluster 12 correlates with increasing TSNE-2 scores.

Similar high percentages of WT and *Cd47⁻/⁻* cells within cluster 12 expressed the erythroid lineage markers Klf1, Aqp1, Epor, Ermap, and Gata1, but percentages for Klf1, Epor, and Gata1 were lower in *Thbs1⁻/⁻* cells (*Figure 4—source data 1*). Expression of the erythroid genes *Klf1, Aqp1, Epor, Ermap, and Gata1* was not detected in the stem cell cluster 14 (*Figure 4—source data 1*). The average

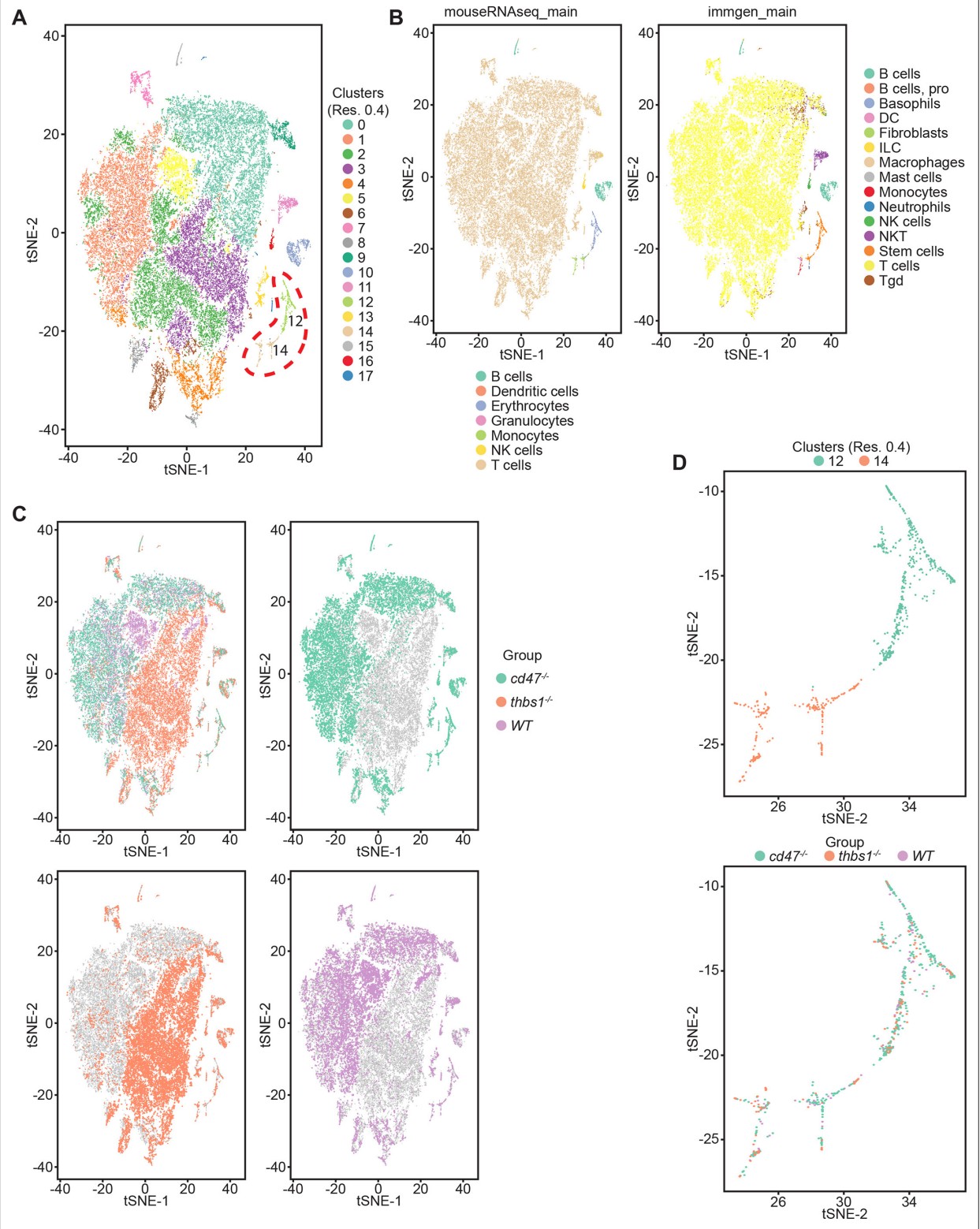

**Figure 3.** Effects of *Cd47* and *Thbs1* gene deletion on stem cell and erythroid precursor populations in mouse spleen identified using single cell RNA sequence analysis. (**A**) tSNE clustering analysis of lineage-depleted spleen cells from WT, *Cd47*$^{-/-}$ and *Thbs1*$^{-/-}$mice (n = 3). The encircled area contains erythroid cells (clusters 12) and stem cells (cluster 14). (**B**) Cell type analysis using Immgen and Mouse RNAseq and SingleR (v.1.0) databases. (**C**) Distribution of WT, *Cd47*$^{-/-}$ and *Thbs1*$^{-/-}$ spleen cells in each cluster of the tSNE plot. (**D**) Enlarged plots of clusters 12 and 14 and cells in these clusters colored by genotype.

*Figure 3 continued on next page*

*Figure 3 continued*

The online version of this article includes the following figure supplement(s) for figure 3:

**Figure supplement 1.** Single cell RNA sequence post-filter QC plots.

mRNA/cell for Tfrc and Ermap was significantly higher in *Cd47⁻/⁻* cells, whereas mRNAs encoding Klf1, Aqp1, Epor, and Gata1 were significantly lower in *Thbs1⁻/⁻* cells compared to WT (*Figure 5A*, *Table 2*). Violin plots indicated that cells with the highest Mki67 mRNA expression were more abundant in *Cd47⁻/⁻* cells in cluster 12, and the average expression was higher (p=9.2 × 10⁻⁶), but cells with high and low Mki67 had similar distributions in *Thbs1⁻/⁻* and WT cells (*Figure 5A*). Mki67 expression levels were lower in cluster 14, and *Cd47⁻/⁻* cells had higher mean expression that WT, but *Thbs1⁻/⁻* cells had lower Mki67 expression than WT (*Table 2*). Expression of Kit was also higher in *Cd47⁻/⁻* versus WT cells in cluster 12 (p=0.0015). Although more *Cd47⁻/⁻* cells expressed Gata1 (*Figure 4—source data 1*), its mean expression was not higher than in WT cells.

Expression of Xpo1 and Ranbp2, which regulates activity of the Xpo1/Ran complex that stabilizes Gata1 (*Ritterhoff et al., 2016*), was higher in cluster 12 than in cluster 14, and within cluster 12 the distribution of positive cells was similar to that for other markers of committed erythroid precursors (*Figure 4*). Xpo1 and RanBP2 mRNAs were expressed in higher percentages of cluster 12 *Cd47⁻/⁻* and *Thbs1⁻/⁻* cells compared to WT (*Figure 4—source data 1*). The average Xpo1 expression/cell in cluster 12 was significantly higher in *Cd47⁻/⁻* and *Thbs1⁻/⁻* cells compared to WT, whereas expression was not *Cd47*- or *Thbs1*-dependent in cluster 14 (*Figure 5A and B*, *Table 2*).

Resolution of cluster 12 into CD34⁺ and CD34⁻ cells revealed that the CD47-dependent expression of Xpo1 and Ranbp2 was restricted to the CD34⁻ population (*Figure 4—source data 2*). Xpo1 was also *Thbs1*-dependent in the CD34⁻ cells but could not be evaluated in CD34⁺ due to lack of an adequate *Thbs1⁻/⁻* cell number. In contrast, expression of mRNA for RanBP1, which regulates the physical interaction of Xpo1 with CD47 (*Kaur et al., 2022*), did not differ in *Cd47⁻/⁻* or *Thbs1⁻/⁻* cells in clusters 12 (*Table 2*).

Nr3c1, a marker of adult definitive erythropoiesis (*Kingsley et al., 2013*), and Ddx46, which is required for hematopoietic stem cell differentiation (*Hirabayashi et al., 2013*), were among the genes with increased percentages of positive cells and significantly increased average expression in both *Cd47⁻/⁻* and *Thbs1⁻/⁻* cells in cluster 12 (*Figure 4—source data 1*, *Table 2*). These genes also showed CD47-dependent expression in cluster 14, but with less significant p-values for expression per cell (*Table 2*).

The co-expression of mRNAs for erythropoietic genes was compared in cluster 12 cells from *Cd47⁻/⁻*, WT, and *Thbs1⁻/⁻* spleens (*Table 3*). Small fractions of the WT CD34⁺ cells expressed Gata1 and Klf1 mRNAs, 2.2% expressed Mki67, and none expressed Ermap. Consistent with the flow cytometry data in *Figure 2*, coexpression of the respective genes with CD34 was more frequent in *Cd47⁻/⁻* cells in cluster 12, but generally less in *Thbs1⁻/⁻* cells. Notably, CD34⁺ *Thbs1⁻/⁻* cells showed no Mki67 coexpression. In contrast, WT cells that expressed committed erythroid differential markers were highly proliferative. Coexpression of all these genes with Mik67 was more frequent in *Cd47⁻/⁻* cells compared to WT cells, but only for Kit and Ly6a in *Thbs1⁻/⁻* cells. Coexpression of the erythroid transcription factors Gata1 and Klf1 was similarly increased in *Cd47⁻/⁻* cells but decreased in *Thbs1⁻/⁻* cells compared to the WT cells in cluster 12. The latter is consistent with the extended life span of *Thbs1⁻/⁻* RBC (*Wang et al., 2020*). Kit coexpression with the erythroid lineage markers Ermap and Klf1 was moderately dependent on genotype, and Ly6a coexpression with these markers was decreased for *Cd47⁻/⁻* cells in cluster 12.

A report that CD47 mediates transfer of mitochondria from macrophages to early erythroblasts during stress-induced erythropoiesis suggested that erythroid precursors in *Cd47⁻/⁻* mice would have lower expression of mitochondrial chromosome-encoded genes (*Yang et al., 2022*). However, *Mt-Atp8*, *Mt-Nd3*, *Mt-Nd4l*, *Mt-Nd5*, and *Mt-Nd6* were among the most significantly up-regulated genes in *Cd47⁻/⁻* cells in cluster 12 (*Table 4*). Consistent with CD47 and TSP1 regulation of mitochondrial homeostasis in other cell types (*Frazier et al., 2011*; *Kelm et al., 2020*; *Miller et al., 2015*; *Norman-Burgdolf et al., 2020*), expression of *mt-Atp8*, *Mt-Nd4l*, *Mt-Nd5*, and *Mt-Nd6* was also increased in *Thbs1⁻/⁻* cells in cluster 12. However, expression of other mitochondrial-encoded genes was significantly decreased in *Thbs1⁻/⁻* cells in cluster 12.

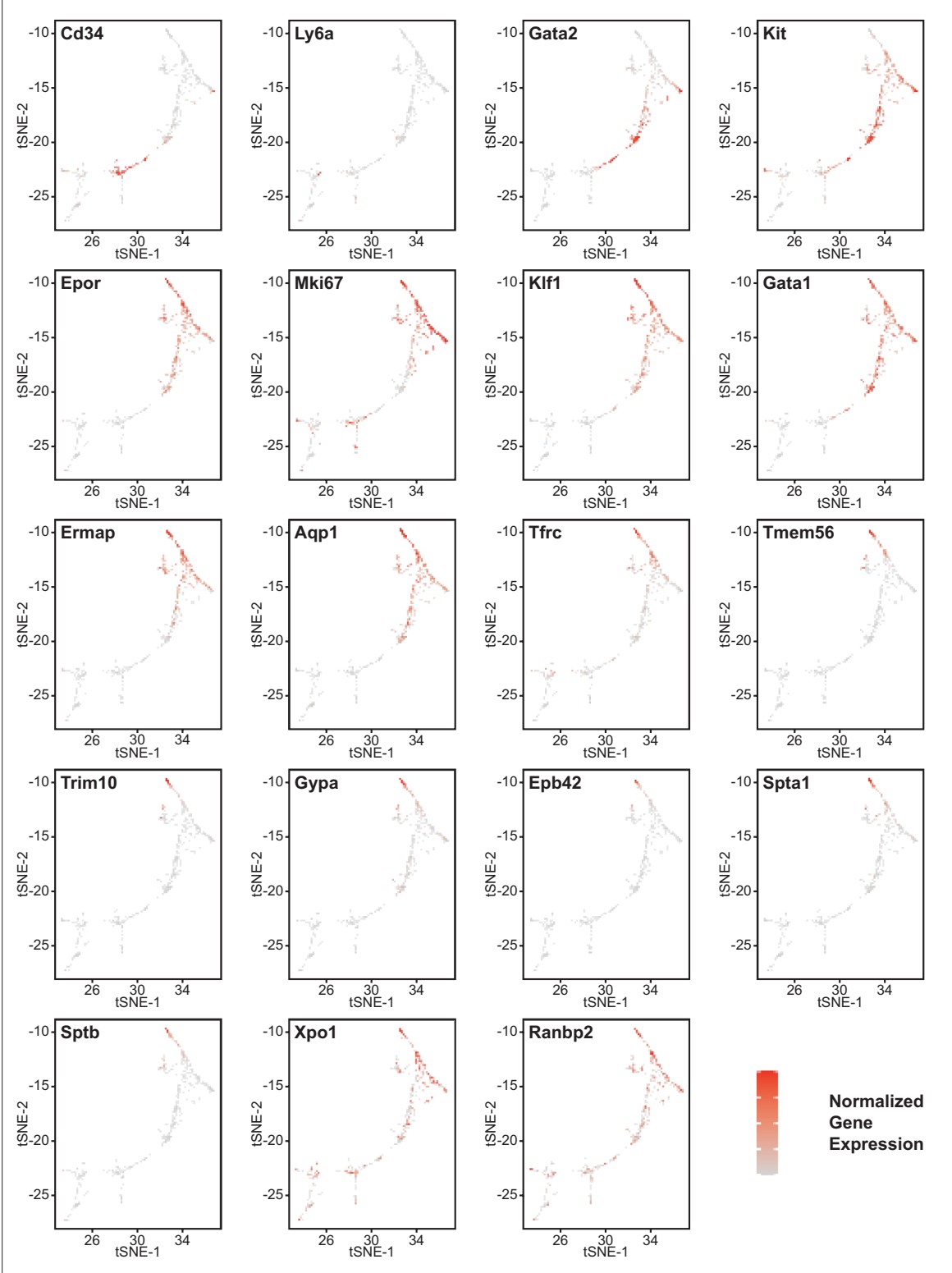

**Figure 4.** Effects of *Cd47* and *Thbs1* gene deletion on gene expression in stem cell and erythroid precursor clusters. High resolution tSNE plots showing the distribution of mRNAs encoding the multipotent stem cell markers CD34 and Ly6a (Sca1) and Gata2, the erythropoietic markers Kit and Epor, the proliferation marker Mki67, erythroid differentiation transcription factors Klf1 and Gata1, and erythroid differentiation and extramedullary erythropoiesis markers Ermap, Aqp1, Tmem56, Trim10, Gypa1, Spta1, Sptb, Ebp42, Xpo1, and RanBP2 in clusters 12 and 14. Expression levels were normalized to maximum expression of each mRNA in these clusters.

*Figure 4 continued on next page*

*Figure 4 continued*

The online version of this article includes the following source data and figure supplement(s) for figure 4:

**Source data 1.** Differential expression of erythropoietic, stem cell, and proliferation associated markers in cell clusters 12 and 14.

**Source data 2.** Differential mRNA expression of the nuclear export protein Xpo1 and nuclear pore protein synthesis instructor Ranbp2 in cluster12 and CD34$^+$ and CD34$^-$ subsets of cluster 12 cells.

**Figure supplement 1.** The distributions of mRNA expression of the indicated genes related to stem cells and erythropoietic cells are shown throughout the 18 clusters as a tSNE projection.

**Figure supplement 2.** Violin plots of erythropoietic, stem cell, and proliferation associated marker mRNAs expressed in 18 cell clusters.

## Reclustering and analysis of cells expressing erythroid signature genes

Erythrocyte progenitor markers were expressed mainly within cluster 12, but some positive cells were scattered across the T cell clusters 0 through 5 (*Figure 6A*, *Figure 6—figure supplement 1A*). To determine whether the latter positive cells include relevant erythroid lineages that were missed in the initial clustering, five lineage markers Gypa, Ermap, Klf1, Gata1, and Aqp1 that predominately localized in cluster 12 (*Figure 6—figure supplement 1A*) were selected as an erythroid signature to calculate module scores (*Figure 6—figure supplement 1A and B*). The 1007 cells that express this signature based on module scores (*Figure 6A*) were then reclustered, yielding five clusters within two major clusters in a UMAP projection (*Figure 6B*). Immgen annotations predicted that the major clusters represent T cells (548 cells) and stem cells (419 cells, *Figure 6C*). Mouse RNAseq annotation confirmed the T cell cluster and predicted the stem cell cluster to be of erythrocyte lineage (*Figure 6C*). WT, *Cd47*$^{-/-}$ and *Thbs1*$^{-/-}$ cells were uniformly distributed throughout the erythroid cluster but segregated within the T cell cluster (*Figure 6D*). *Cd47*$^{-/-}$ cells were more abundant in the erythroid cluster (271 cells) and *Thbs1*$^{-/-}$ cells were less abundant (58 cells) relative to WT cells (90 cells).

A volcano plot indicated major differences in the transcriptomes of the two main clusters consistent with their respective erythroid and T cell lineages (*Figure 6—figure supplement 1C*). The expression of Ermap, Klf1, and Aqp1 mRNAs in 20–30% of cells in the T cell cluster is consistent with previous reports of their expression in minor subsets of T cells (*Moon et al., 2004*; *Su et al., 2021*; *Teruya et al., 2018*; *Figure 7*).

The distribution of erythroid markers throughout the reclustered erythroid population was consistent with the results for cluster 12 (*Figure 7*). CD34$^+$ cells were concentrated in the lower region of the erythroid cell cluster. Consistent with the selection of cells based on expression of committed erythroid lineage genes, the cluster lacked Ly6a$^+$ cells. Kit was expressed by the CD34$^+$ cells and extended upward through the cluster. The upper cells showed increased expression of Epor, Klf1, Gata1, and Aqp1. Expression of the proliferation marker Mki67 was strongest in the upper region of the erythroid cell cluster and extended to cells that cells that expressed markers of more mature precursors including Ermap, Tfrc (transferrin receptor), and Tmem56. Trim10 mediates terminal erythroid differentiation and colocalized with mRNAs encoding glycophorin A, spectrin A, spectrin B, and band 4.2. The upregulation of erythrocyte lineage markers coincided with more abundant *Cd47*$^{-/-}$ cells in this cluster, which supports the initial unsupervised clustering and confirms increased extramedullary erythropoiesis in *Cd47*$^{-/-}$ mice.

Differential gene expression analysis contrasting *Cd47*$^{-/-}$ and *Thbs1*$^{-/-}$ with WT cells in the erythroid cluster (*Table 5*) reproduced all of the results found in cluster 12 (*Table 2*). Mki67, Tfrc, and Ermap mRNAs were significantly higher in *Cd47*$^{-/-}$ cells compared to WT, whereas Klf1, Aqp1, Epor, and Gata1 mRNAs were significantly lower in *Thbs1*$^{-/-}$ cells (*Table 5*). Differences in the percentages of cells expressing these genes seen in cluster 12 were also reproduced in the reclustered erythroid cells (*Figure 7—source data 1*). The altered expression of Xpo1 and RanBP2 were also confirmed in the erythroid cluster, but Xpo1 mRNA expression was not CD47-dependent in the T cell cluster (*Table 5*). Notably, the increased expression of Nr3c1, and Ddx46 mRNAs in *Cd47*$^{-/-}$ and *Thbs1*$^{-/-}$ cells in the erythroid cell cluster was also found in the reclustered T cells (*Table 5*). However, the T cell cluster completely lacked expression of the proliferation-associated markers Mki67 and Tfrc (*Figure 7—source data 1*). Aqp1 and Gata1were also expressed by a subset of the T cells (*Figure 7—source data 1*), but their decreased mRNA expression in *Thbs1*$^{-/-}$ cells was not seen in the T cell cluster (*Table 5*).

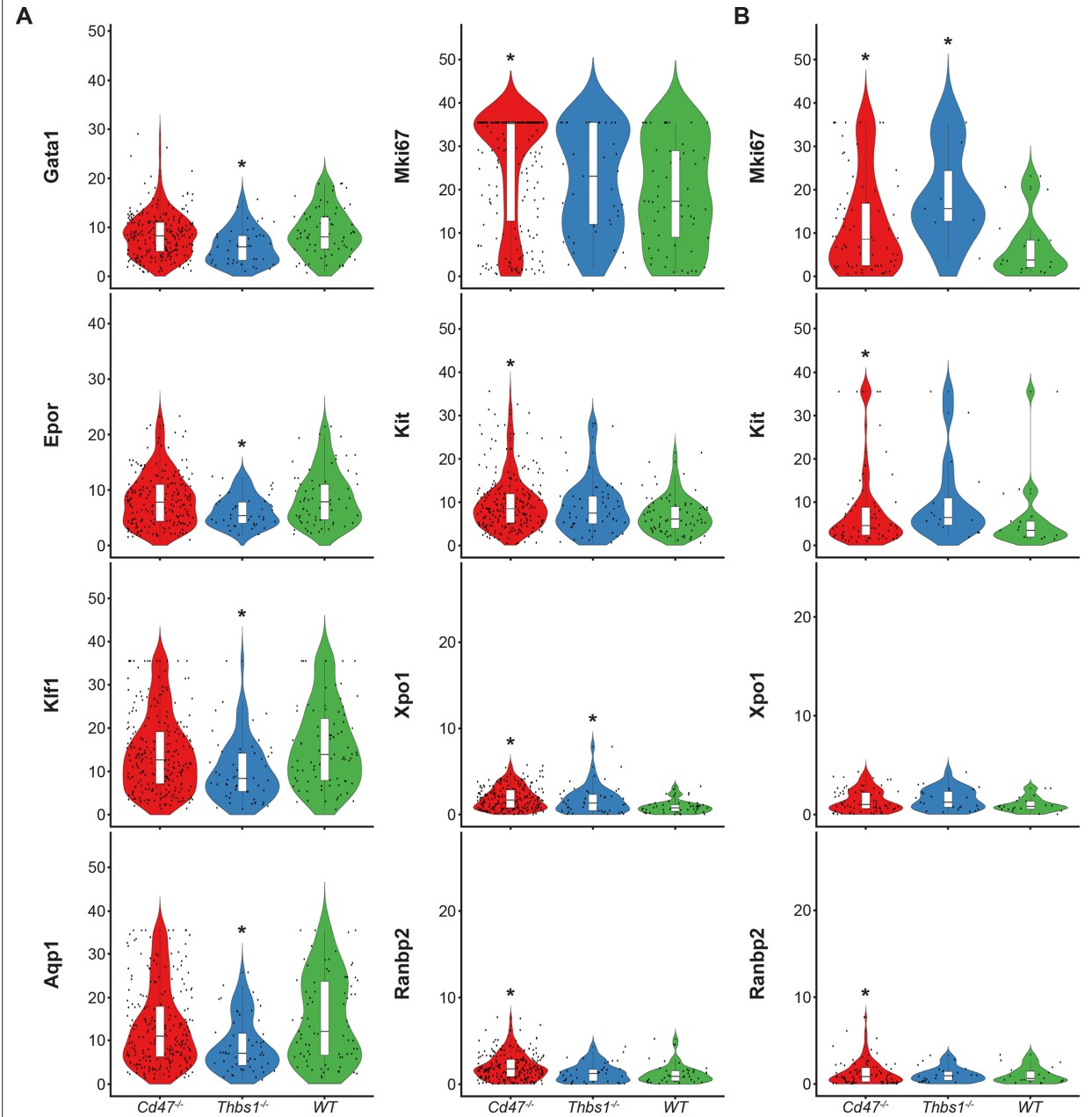

**Figure 5.** Differential effects of *Cd47* and *Thbs1* gene deletion on mRNA expression levels in erythroid precursor and stem cell clusters 12 and 14.
(**A**) Violin plots comparing mRNA expression levels of the indicated genes in *Cd47*⁻/⁻(red), *Thbs1*⁻/⁻(blue) and WT spleen cells (green) in cluster 12.
(**B**) Violin plots comparing mRNA expression levels of the indicated genes in *Cd47*⁻/⁻, *Thbs1*⁻/⁻, and WT spleen cells in cluster 14. *=p < 0.05 relative to WT cells in the respective cluster. n = 2 female and 1 male mice of each genotype; *=p < 0.05.

## Discussion

Our data supports a role for thrombospondin-1-stimulated CD47 signaling to limit early stages of erythropoiesis in WT mouse spleen and opposing roles for thrombospondin-1 and CD47 to regulate SIRPα-dependent turnover of RBC. The increased anemic stress in *Cd47*⁻/⁻ spleen increases extramedullary erythropoiesis, whereas extramedullary erythropoiesis is suppressed by the decreased RBC turnover in *Thbs1*⁻/⁻ spleen. These studies are consistent with the reported increased red cell turnover in *Cd47*⁻/⁻ mice and decreased RBC turnover in *Thbs1*⁻/⁻ mice compared to WT mice (*Wang et al., 2020*). Increased RBC clearance in *Cd47*⁻/⁻ mice is mediated by loss of the 'don't eat me' function of CD47 on red cells (*Kim et al., 2018*; *Oldenborg et al., 2000*). In wildtype mice, clearance is

**Table 2.** Differential mRNA expression of erythropoietic, stem cell, and proliferation associated markers in WT, $Cd47^{-/-}$, and $Thbs1^{-/-}$ cells in clusters 12 and 14.

| Cluster | Gene | $Cd47^{-/-}$ vs WT | | $Thbs1^{-/-}$ vs WT | |
|---|---|---|---|---|---|
| | | p-value | Avg log$_2$ FC | p-value | Avg log$_2$ FC |
| 12 | Klf1 | 0.463 | –0.115 | 0.0023 | –0.510 |
| 12 | Aqp1 | 0.331 | –0.130 | 0.0011 | –0.472 |
| 12 | Tfrc | 0.017 | 0.531 | 0.93 | 0.009 |
| 14 | Tfrc | 0.64 | 0.071 | 0.922 | 0.016 |
| 12 | Epor | 0.506 | –0.078 | $4.75\times10^{-5}$ | –0.576 |
| 12 | Ermap | $5.1\times10^{-3}$ | 0.309 | 0.65 | –0.073 |
| 12 | Gata1 | 0.57 | 0.007 | 0.0011 | –0.377 |
| 12 | Mki67 | $9.16\times10^{-6}$ | 0.997 | 0.22 | 0.456 |
| 14 | Mki67 | 0.0089 | 0.798 | 0.0017 | –0.062 |
| 12 | Kit | 0.0015 | 0.331 | 0.057 | 0.264 |
| 14 | Kit | 0.020 | 0.409 | 0.58 | 0.222 |
| 12 | Xpo1 | $3.31\times10^{-8}$ | 0.472 | 0.0069 | 0.323 |
| 14 | Xpo1 | 0.079 | 0.211 | 0.19 | 0.148 |
| 12 | Ranbp1 | 0.24 | 0.108 | 0.13 | 0.079 |
| 14 | Ranbp1 | 0.91 | 0.069 | 0.076 | –0.486 |
| 12 | Ranbp2 | $1.98\times10^{-14}$ | 0.754 | 0.092 | 0.269 |
| 14 | Ranbp2 | 0.0056 | 0.365 | 0.88 | 0.042 |
| 12 | Nr3c1 | $8.5\times10^{-4}$ | 0.320 | $8.8\times10^{-4}$ | 0.430 |
| 14 | Nr3c1 | 0.012 | 0.153 | $8.6\times10^{-5}$ | 0.428 |
| 12 | Ddx46 | $3.04\times10^{-8}$ | 0.530 | $2.68\times10^{-10}$ | 0.779 |
| 14 | Ddx46 | 0.0014 | 0.385 | 0.0023 | 0.490 |

augmented by thrombospondin-1 binding to the clustered CD47 on aging red cells (**Wang et al., 2020**). Thus, anemic stress in the mouse strains studied here and the abundance of committed erythroid progenitors in their spleens decrease in the order $Cd47^{-/-}$>WT > $Thbs1^{-/-}$.

Independent of CD47 protecting erythropoietic cells from phagocytosis, we previously found that bone marrow from $Cd47^{-/-}$ mice subjected to the stress of ionizing radiation exhibited more colony forming units for erythroid (CFU-E) and burst-forming unit-erythroid (BFU-E) progenitors compared to bone marrow from irradiated wildtype mice (**Maxhimer et al., 2009**). Loss of CD47 results in an intrinsic protection of hematopoietic stem cells in bone marrow from genotoxic stress, which may be mediated by an increased protective autophagy response (**Soto-Pantoja et al., 2012**). The same mechanism may contribute to regulating extramedullary erythropoiesis in spleen.

The properties of erythroid precursors that accumulate in $Cd47^{-/-}$ spleens are consistent with previous studies of stress induced extramedullary erythropoiesis associated with malaria or trypanosome infections (**Delic et al., 2020**; **Thompson et al., 2010**). In addition to containing elevated NK precursors (**Nath et al., 2018**), the present data demonstrate that $Cd47^{-/-}$ spleen contains more abundant erythroid precursors that are Ter119$^+$ by flow cytometry but presumably lack sufficient Ter119 for antibody bead depletion. These cells are present but less abundant in WT spleen, indicating that a low basal level of extramedullary erythropoiesis occurs in healthy mouse spleen. The Ter119$^+$CD34$^-$ cells that accumulate in $Cd47^{-/-}$ spleens are more proliferative and express multiple markers of committed erythroid precursors. In contrast, the same cells are depleted in $Thbs1^{-/-}$ spleen, consistent with the function of thrombospondin-1 to facilitate the CD47/SIRPα-mediated turnover of aging RBC (**Wang**

**Table 3.** Co-expression of the indicated erythropoiesis related genes in WT, *Cd47⁻/⁻* and *Thbs1⁻/⁻* spleen cells in cluster 12 was quantified and expressed as a percentage of the total cell number of each genotype in cluster 12.

| Cluster 12 gene coexpression | WT | Cd47⁻/⁻ | Thbs1⁻/⁻ |
|---|---|---|---|
| *Cd34_Ermap* | 0.0% | 1.8% | 1.6% |
| *Cd34_Klf1* | 6.5% | 8.1% | 3.1% |
| *Cd34_Gata1* | 6.5% | 8.4% | 0.0% |
| *Mki67_Cd34* | 2.2% | 6.3% | 0.0% |
| *Mki67_Ermap* | 55.4% | 63.0% | 54.7% |
| *Mki67_Kit* | 48.9% | 67.2% | 57.8% |
| *Mki67_Ly6a* | 37.0% | 48.2% | 43.8% |
| *Mki67_Klf1* | 62.0% | 74.6% | 57.8% |
| *Mki67_Epor* | 54.4% | 62.7% | 53.1% |
| *Klf1_Ermap* | 67.4% | 68.0% | 62.5% |
| *Klf1_Gata1* | 78.3% | 82.4% | 59.4% |
| *Ermap_Gata1* | 62.0% | 65.1% | 56.2% |
| *Kit_Ermap* | 54.4% | 56.7% | 59.4% |
| *Kit_Klf1* | 71.7% | 77.1% | 78.1% |
| *Ly6a_Ermap* | 42.4% | 37.7% | 46.9% |
| *Ly6a_Klf1* | 64.1% | 59.2% | 67.2% |

*et al., 2020*). These data also indicate that physiological levels of thrombospondin-1 are necessary to support a basal level of erythropoiesis in WT spleen.

Notably, *Thbs1⁻/⁻* and *Cd47⁻/⁻* spleens contain more early erythroid precursors than are maintained basally in a WT spleen, consistent with the role of thrombospondin-1 signaling via CD47 to limit the expression of multipotent stem cell transcription factors in spleen (*Kaur et al., 2013*). Earlier erythroid precursors that are Ter119⁻Kit⁺ accumulate in *Thbs1⁻/⁻* spleen. *Thbs1⁻/⁻* and *Cd47⁻/⁻* cells express more Ddx46, which is required for differentiation of hematopoietic stem cells (*Hirabayashi et al., 2013*), and Xpo1, which supports the erythropoietic function of Gata1 (*Guillem et al., 2020*). Although Ter119⁺ cells expressing markers of committed erythroid progenitors were depleted in *Thbs1⁻/⁻* compared to WT spleens, mRNAs for some markers of committed erythroid cells including Nr3c1 mRNA were elevated in *Thbs1⁻/⁻* and *Cd47⁻/⁻* cluster 12 cells. However, early progenitors express CD45R, and inclusion of this antibody in the negative selection cocktail should deplete early progenitors from the populations used for both RNAseq analyses. This may account for the relative lack of CD47-dependent CD34⁺ cells in cluster 12.

One caveat in interpreting the CD47- and thrombospondin-1-dependence of the

**Table 4.** Differential expression of mitochondrial encoded genes in cluster 12 cells from WT, *Cd47⁻/⁻*, and *Thbs1⁻/⁻* spleens.

| Gene | p-val Cd47⁻/⁻ vs WT | Avg log₂FC Cd47⁻/⁻ vs_WT | p-val Thbs1⁻/⁻ vs_WT | Avg log₂FC Thbs1⁻/⁻ vs_WT |
|---|---|---|---|---|
| *Mt-Atp6* | 0.887 | −0.027 | 5.43×10⁻¹¹ | −0.812 |
| *Mt-Atp8* | 7.23×10⁻²⁰ | 0.98 | 0.0168 | 0.377 |
| *Mt-Co1* | 0.028 | 0.135 | 2.60×10⁻⁷ | −0.532 |
| *Mt-Co2* | 0.226 | 0.068 | 3.37×10⁻¹⁰ | −0.693 |
| *Mt-Co3* | 0.207 | 0.097 | 1.02×10⁻⁸ | −0.671 |
| *Mt-Cytb* | 0.338 | −0.006 | 9.35×10⁻¹⁰ | −0.790 |
| *Mt-Nd1* | 0.074 | 0.15 | 6.78×10⁻⁶ | −0.636 |
| *Mt-Nd2* | 0.135 | 0.113 | 1.00×10⁻⁷ | −0.748 |
| *Mt-Nd3* | 7.67×10⁻⁹ | 0.531 | 1.36×10⁻⁷ | −0.813 |
| *Mt-Nd4* | 0.152 | 0.093 | 5.84×10⁻⁶ | −0.558 |
| *Mt-Nd4l* | 3.50×10⁻¹⁰ | 0.658 | 6.49×10⁻⁸ | 0.787 |
| *Mt-Nd5* | 4.63×10⁻⁸ | 0.612 | 0.0154 | 0.375 |
| *Mt-Nd6* | 1.65×10⁻⁷ | 0.416 | 4.39×10⁻⁶ | 0.524 |

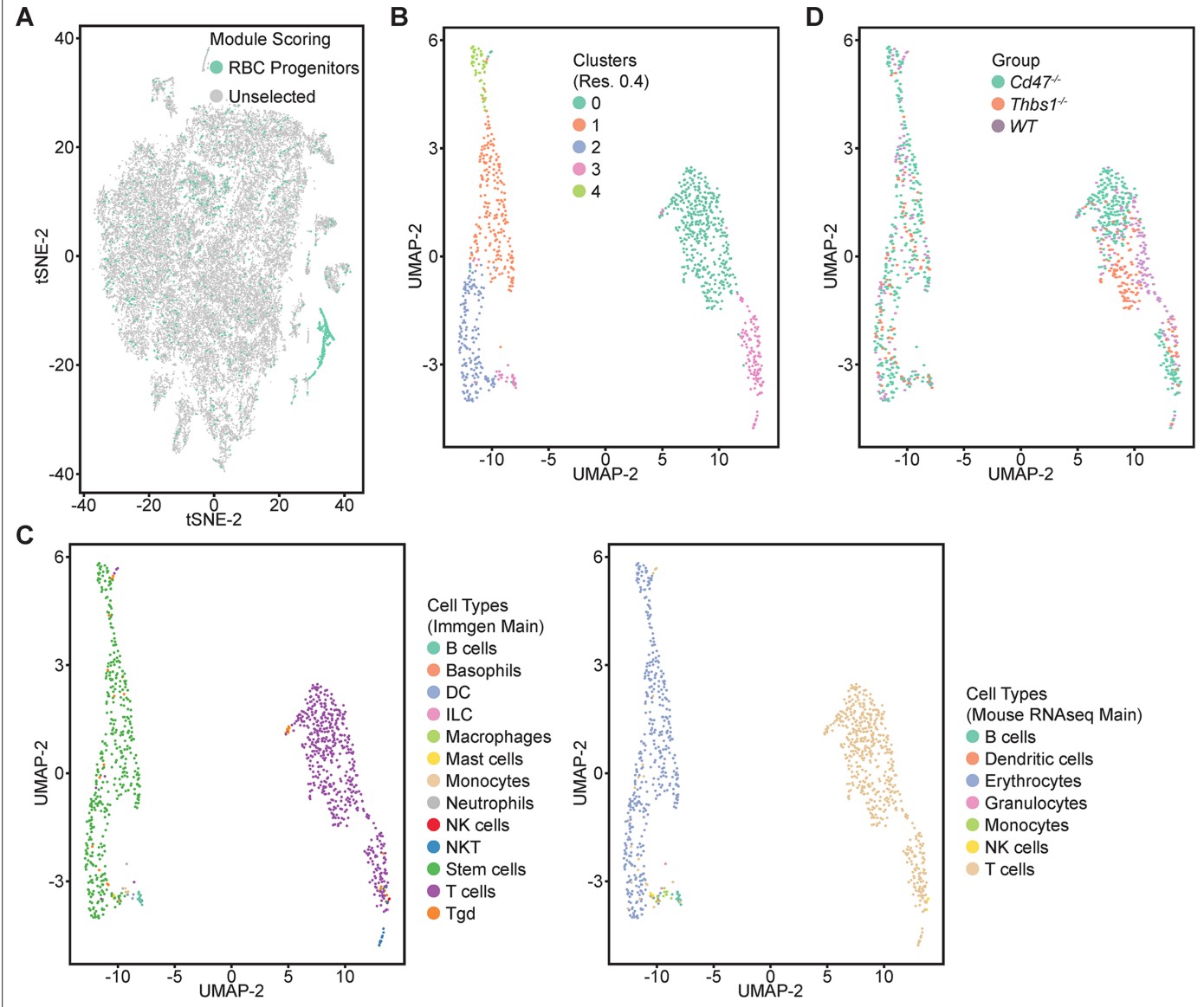

**Figure 6.** Re-clustering of lineage-depleted spleen cells selected for expression of erythroid signature genes. (**A**) tSNE plot showing the distribution of cells selected for expressing threshold levels of Gypa, Ermap, Klf1, Gata1, and/or Aqp1. (**B**) Re-clustered cells expressing the erythroid progenitor signature are displayed in an UMAP projection. (**C**) Immgen and mouse RNAseq main cell type annotation of reclustered cells expressing the erythroid gene signature. (**D**) Distribution of WT (purple), *Cd47⁻/⁻* (green), and *Thbs1⁻/⁻* cells (orange) in the erythroid precursor and T cell clusters. Data are from two female and one male mouse of each genotype.

The online version of this article includes the following figure supplement(s) for figure 6:

**Figure supplement 1.** Strategy for reclustering spleen cells that express a gene signature for committed erythroid precursors.

extramedullary erythropoiesis markers Ermap and Aqp1 in total spleen cells and the Ter119-depleted cells used for scRNAseq is that both are also expressed in minor subsets of T cells (**Moon et al., 2004**). The reclustering analysis confirmed that these erythropoietic markers are expressed in minor T cell populations, which notably also showed CD47-dependent gene expression changes. This may also account for the differences in CD47-dependence of erythropoietic marker expression observed by flow cytometry in Ter119⁺ cells but not in total spleen cells or Ter119⁻ spleen cells.

In addition to utility for assessing extramedullary erythropoiesis, the CD47-dependent erythropoiesis genes identified here may have translational utility. Therapeutic CD47 antibodies and decoys designed to inhibit the function of CD47 have entered multiple clinical trials for treating cancers, but anemia associated with loss of CD47-dependent inhibitory SIRPα signaling in macrophages has been

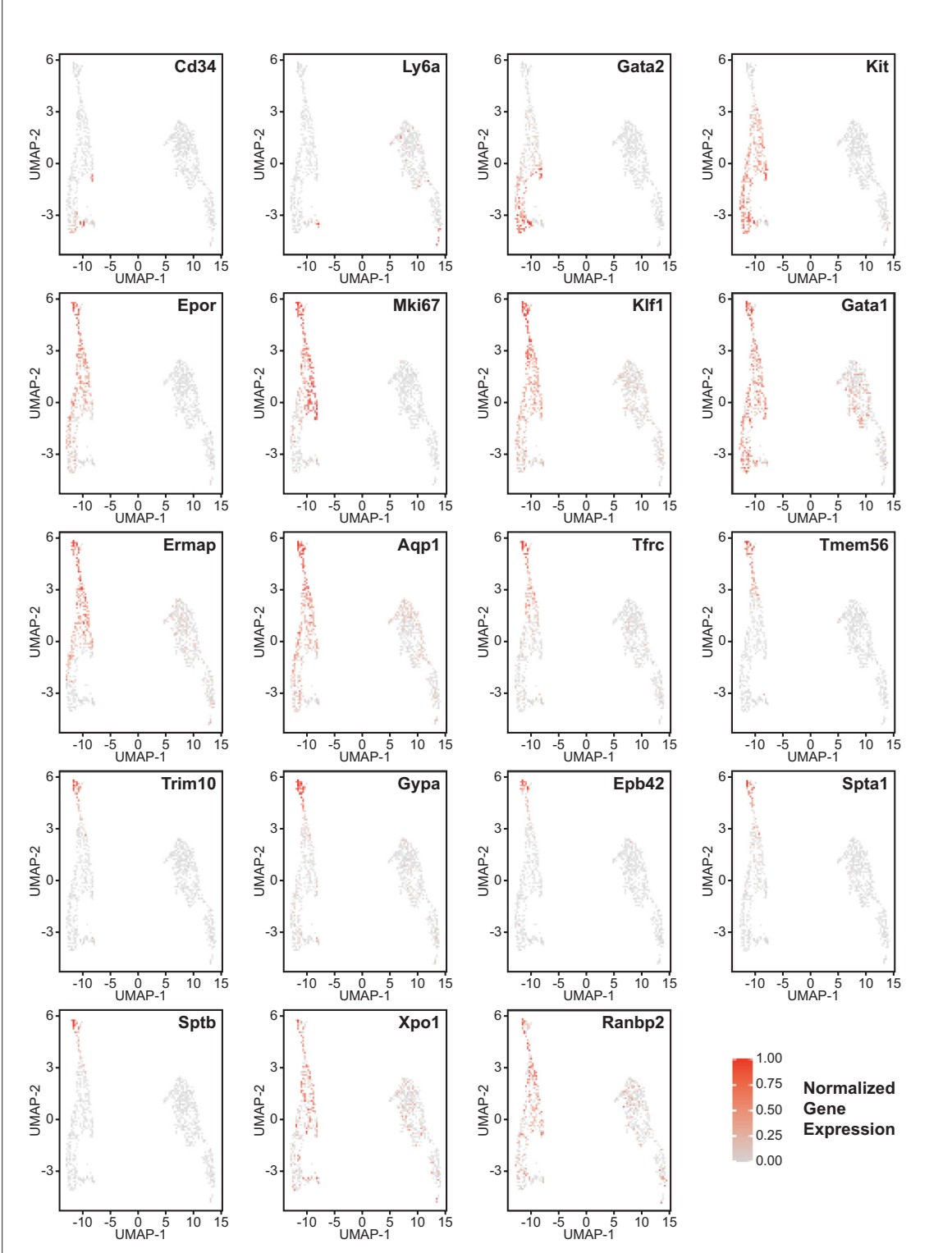

**Figure 7.** Gene expression in reclustered lineage-depleted spleen cells selected for expression of erythroid signature genes. Distribution of the multipotent stem cell markers CD34 and Ly6a (Sca1), erythropoietic markers Gata2, Kit, and Epor, the erythroid differentiation transcription factors Klf1 and Gata1, and erythroid differentiation and extramedullary erythropoiesis marker the proliferation marker Mik67, and the erythroid markers Ermap, Tfrc, Aqp1, Tfrc, Tmem56, Trim10, Gypa1, Ebp42, Spta1, Sptb, Xpo1, and Ranbp2 in the erythroid lineage cluster (left) and T cell cluster (right). Expression levels were normalized to maximum expression of each mRNA in these clusters. Data are from two female and one male mouse of each genotype.

*Figure 7 continued on next page*

*Figure 7 continued*

The online version of this article includes the following source data for figure 7:

**Source data 1.** Differential expression of erythropoietic, stem cell, and proliferation associated markers in reclustered erythroid and T cell clusters.

a frequent side effect observed for the first generation of these therapeutics (*Kaur et al., 2020*). These therapeutic antibodies also recognize CD47 on RBC and thereby sensitize them to removal by phagocytes. The resulting anemia would be expected to induce erythropoiesis in bone marrow and possibly at extramedullary sites. Human spleen cells are not accessible to directly evaluate extra-medullary erythropoiesis in cancer patients receiving CD47-targeted therapeutics, but analysis of circulating erythroid precursors or liquid biopsy methods could be useful to detect induction of extra-medullary erythropoiesis by these therapeutics. Therefore, the CD47-dependent erythroid markers

**Table 5.** Differential mRNA expression of erythropoietic, stem cell, and proliferation associated markers in reclustered WT, $Cd47^{-/-}$, and $Thbs1^{-/-}$ erythroid and T cell clusters.

| Cluster | Gene | $Cd47^{-/-}$ vs WT | | $Thbs1^{-/-}$ vs WT | |
| --- | --- | --- | --- | --- | --- |
| | | p-value | Avg log$_2$ FC | p-value | Avg log$_2$ FC |
| Erythroid | Klf1 | 0.507 | –0.095 | 0.0016 | –0.526 |
| Erythroid | Aqp1 | 0.196 | –0.088 | 8.1x10$^{-4}$ | –0.390 |
| T cells | Aqp1 | 0.662 | 0.027 | 0.350 | –0.092 |
| Erythroid | Tfrc | 0.015 | 0.549 | 0.936 | 0.054 |
| T cells | Tfrc | - | - | - | - |
| Erythroid | Epor | 0.767 | –0.046 | 0.026 | –0.219 |
| Erythroid | Ermap | 0.0064 | 0.326 | 0.496 | –0.010 |
| T cells | Ermap | 0.335 | 0.076 | 0.697 | 0.063 |
| Erythroid | Gata1 | 0.38 | 0.009 | 0.0040 | –0.337 |
| T cells | Gata1 | 0.0044 | –.162 | 0.302 | 0.093 |
| Erythroid | Mki67 | 4.6x10$^{-6}$ | 1.015 | 0.118 | 0.545 |
| T cells | Mki67 | - | - | - | - |
| Erythroid | Kit | 0.0066 | 0.293 | 0.059 | 0.275 |
| T cells | Kit | - | - | - | - |
| Erythroid | Xpo1 | 2.65x10$^{-9}$ | 0.514 | 0.0025 | 0.373 |
| T cells | Xpo1 | 0.238 | 0.047 | 0.137 | 0.066 |
| Erythroid | Ranbp1 | 0.099 | 0.133 | 0.41 | 0.078 |
| T cells | Ranbp1 | 0.278 | –0.124 | 0.970 | 0.019 |
| Erythroid | Ranbp2 | 3.2x10$^{-14}$ | 0.755 | 0.058 | 0.298 |
| T cells | Ranbp2 | 4.5x10$^{-4}$ | 0.275 | 0.210 | 0.094 |
| Erythroid | Nr3c1 | 6.5x10$^{-4}$ | 0.319 | 0.0011 | 0.447 |
| T cells | Nr3c1 | 0.0018 | 0.274 | 0.0018 | 0.233 |
| Erythroid | Ddx46 | 1.65x10$^{-9}$ | 0.530 | 2.98x10$^{-9}$ | 0.775 |
| T cells | Ddx46 | 2.3x10$^{-5}$ | 0.348 | 7.5x10$^{-6}$ | 0.446 |
| Erythroid | Hba-a1 | 0.906 | 0.034 | 0.0088 | –2.168 |

identified in this study may be useful biomarkers for assessing hematologic side effects of CD47-targeted therapeutics.

# Materials and methods

## Key resources table

| Reagent type (species) or resource | Designation | Source or reference | Identifiers | Additional information |
|---|---|---|---|---|
| Commercial assay or kit | CD8a+T Cell Isolation Kit | Miltenyi Biotec | Cat#: 130–104–075 | |
| Commercial assay or kit | CD8a (Ly-2) MicroBeads mouse | Miltenyi Biotec | Cat#: 130-117-044 | |
| Chemical compound, drug | EDTA solution | Sigma-Aldrich | Cat#: E8008 | 2 mM |
| Peptide, recombinant protein | bovine serum albumin (BSA) | Sigma-Aldrich | Cat#: A7906 | 0.5% |
| Chemical compound, drug | ACK Lysing Buffer, 100 mL | Quality Biologicals | Cat#: 118-156-721 | |
| Strain, strain background (*Mus musculus*, C57BL/6) | WT mice | Jackson Laboratories | WT C57BL/6 | |
| Strain, strain background (*M. musculus*, C57BL/6) | *Cd47$^{-/-}$* mice | Jackson Laboratories | B6.129S7-*Cd47$^{tm1Fpl}$*/J; Strain:003173 | *Lindberg et al., 1996*; PMID:8864123 |
| Strain, strain background (*M. musculus*, C57BL/6) | *Thbs1$^{-/-}$* mice | Jackson Laboratories | B6.129S2-*Thbs1$^{tm1Hyn}$*/J; Strain:006141 | *Lawler et al., 1998*; PMID:9486968 |
| Antibody | Anti-mouse Ter119-APC (Rat monoclonal) | Biolegend | Cat#: 116212; RRID:AB_313713 | IgG2b FACS (1 µg/100 µl/1 million cells) |
| Antibody | anti-mouse CD34-Percp/Cy5.5 (Rat monoclonal) | Biolegend | Cat#: 119327; RRID:AB_2728136 | IgG2a FACS (1 µg/100 µl/1 million cells) |
| Antibody | anti mouse Sca1 PE/Cy7 (Rat monoclonal) | Biolegend | Cat#: 108114; RRID:AB_493596 | IgG2a FACS (0.5 µg/100 µl/1 million cells) |
| Antibody | anti-mouse cKit PE (Rat monoclonal) | Biolegend | Cat#: 105808; RRID:AB_313217 | IgG2b FACS (0.1 µg/100 µl/1 million cells) |
| Antibody | anti-mouse Ki67- PE/Cy7 (Rat monoclonal) | Biolegend | Cat#: 652425; RRID:AB_2632693 | IgG2a FACS (0.5 µg/100 µl/1 million cells) |
| Antibody | IgG2b, κ APC Isotype Control Antibody (Rat monoclonal) | Biolegend | Cat#: 400611; RRID:AB_326555 | IgG2b FACS (1 µg/100 µl/1 million cells) |
| Antibody | IgG2a, κ PerCP/Cyanine5.5 Isotype Control Antibody (Rat monoclonal) | Biolegend | Cat#: 400531; RRID:AB_2864286 | FACS (1 µg/100 µl/1 million cells) |
| Antibody | IgG2a, κ PE/Cyanine7 Isotype Control Antibody (Rat monoclonal) | Biolegend | Cat#: 400521; RRID:AB_326542 | FACS (0.25 µg/100 µl/1 million cells) |
| Antibody | IgG2b, κ PE Isotype Control Antibody (Rat monoclonal) | Biolegend | Cat#: 400608; RRID:AB_326552 | FACS (0.1 µg/100 µl/1 million cells) |
| Antibody | IgG2a, κ Alexa Fluor 488 Isotype Control Antibody (Rat monoclonal) | Biolegend | Cat#: 400525; RRID:AB_2864283 | FACS (0.25 µg/100 µl/1 million cells) |
| Antibody | anti-Rabbit IgG (H+L) Cross-Adsorbed, Alexa Fluor 594 (Goat polyclonal) | Thermo Fisher | Cat#: A-11012 | FACS (0.2 µg/100 µl/1 million cells) |
| Antibody | Anti-ERMAP (Rabbit polyclonal) | Thermo Fisher | Cat#: BS-12333R | FACS (1:100 dilution) |
| Antibody | Anti-GYPA (Rabbit polyclonal) | Thermo Fisher | Cat#: BS-2575R | FACS (1:100 dilution) |

*Continued on next page*

*Continued*

| Reagent type (species) or resource | Designation | Source or reference | Identifiers | Additional information |
|---|---|---|---|---|
| Antibody | Anti-Aquaporin 1 (Rabbit polyclonal) | Thermo Fisher | Cat#: PA5-78806 | FACS (1 µg/100 µl/1 million cells) |
| Antibody | Anti-EPOR (Rabbit polyclonal) | Bioss; Thermo Fisher | Cat#: BS-1424R | FACS (1 µg/100 µl/1 million cells) |

## Mice and cells

WT, *Cd47$^{-/-}$* (B6.129S7-Cd47tm1Fpl/J, *Lindberg et al., 1996*), and *Thbs1$^{-/-}$* (B6.129S2-*Thbs1$^{tm1Hyn}$*/J, *Lawler et al., 1998*) mice were obtained from The Jackson Laboratory, backcrossed on the C57BL/6 background, and maintained under specific pathogen free conditions. All animal experiments were carried out in strict accordance with the Recommendations for the Care and Use of Laboratory Animals of the National Institutes of Health under a protocol approved by the NCI Animal Care and Use Committee (LP-012). Age and gender matched 8- to 12-week-old mice were used for experiments except where noted. For the bulk RNA sequencing four male *Cd47$^{-/-}$* mice and four male wildtype mice were used per CCBR guidance. For single-cell RNA sequencing, two female and one male of each genotype were used per CCBR guidance. For flow cytometry, two female and two males of each genotype were used.

Spleens were removed from the mice and homogenized in HBSS and passed through a 70 µm mesh (Sigma, CLS431751) to remove debris. The cell suspension was treated with ACK lysis buffer for 4 min to lyse the RBC. The suspensions were centrifuged and washed twice with cold HBSS. Aliquots of single-cell suspensions were stained using trypan blue and counted to assess viability.

## Flow cytometry

Spleens were obtained from two male and two female mice of each genotype. Single-cell suspensions from spleens were stained by incubation for 30 min at 4 °C using optimized concentrations of antibodies: CD34-Percp/Cy5.5 and Ter119-APC, cKit-PE and PE/Cy7, Sca1- AF488, Ki67- PE-Cy7 and Percp/Cy5.5 (Biolegend). Non-tagged Ermap, Gypa, Aqp1 (Thermo Fisher) and Epor (Bios Inc) antibodies were detected using secondary goat anti-rabbit AF594 antibodies. Stained single-cell suspensions were acquired on an LSRFortessa SORP (BD Biosciences), and data were analyzed using FlowJo software (Tree Star). A total of $2 \times 10^5$ gated live events were acquired for each analysis. Isotype and unstained controls were used to gate the desired positive populations.

## Single-cell RNA sequencing (scRNAseq)

Because bulk RNA sequencing analysis identified elevated expression of erythropoietic genes in CD8$^+$ spleen cells from *Cd47$^{-/-}$* mice that were obtained using magnetic bead depletion of all other lineages, the same method was used as the first step to enrich erythroid precursors. Single cell suspensions from WT, *Thbs1$^{-/-}$* and *Cd47$^{-/-}$* spleens were depleted of all mature hematopoietic cell lineages including erythroblasts and mature RBC using the CD8a+T Cell Isolation Kit, mouse (Miltenyi Biotec). Single-cell suspensions were incubated with the supplied antibody cocktail of the CD8+ T cell isolation kit for 15 min on ice and then passed through the column as per manufacturer's instructions. The flow through combined with three washes was centrifuged, and the cells were then incubated with CD8a (Ly-2) microbeads and passed through magnetic columns to obtain lineage-depleted cell populations. Capture & Library Preparation for single cell end-counting gene expression using the 10 X Genomics platform was performed by the Single Cell Analysis Facility (CCR).

Single-cell RNA-sequencing (scRNA-seq) data were generated using the Chromium Single Cell 3′ Solution (10 x Genomics). Raw sequencing data were processed using the CellRanger software suite (version 4.0.0, 10 x Genomics). CellRanger's mkfastq and count functions were utilized to demultiplex raw base call (BCL) files into sample-specific FASTQ files, perform barcode processing, and align cDNA reads to the reference genome (mm10) to generate feature-barcode matrices.

Downstream analysis and visualization were performed within the NIH Integrated Analysis Platform (NIDAP) using R programs developed by a team of NCI bioinformaticians on the Foundry platform (Palantir Technologies). The Single Cell workflow on NIDAP executes the SCWorkflow package (https://github.com/NIDAP-Community/SCWorkflow; copy archived at *NIDAP-Community, 2024*), which is based on the Seurat workflow (v. 4.1.1; *Hao et al., 2021*). Quality control metrics were assessed to

remove low-quality cells, defined as cells with fewer than 200 detected genes or a high percentage (>15%) of mitochondrial gene expression, indicative of cellular stress or apoptosis. After filtering, the dataset consisted of 13,933 single cells with an average of 7523 unique molecular identifiers (UMIs) per cell and an average gene detection of 2114 genes per cell. The dataset underwent normalization via log-transformation, and RunPCA applied to the scaled data of the variable features to compute the principal components. After performing SCTransform (*Hafemeister and Satija, 2019*) on the merged dataset, an unsupervised clustering methodology was employed to categorize cells exhibiting analogous expression patterns. The FindClusters function was used with a clustering resolution of 0.4 for all cells and a reclustered resolution of 0.2 for the RBC progenitor cell subset. Each cellular cluster was annotated with the expression of established cell-type-specific marker genes and dimensionality reduction plots (t-distributed stochastic neighbor embedding [t-SNE] and uniform manifold approximation and projection [UMAP]), were utilized for the visual representation of clusters. Cell types were called using SingleR (v.1.0) (*Aran et al., 2019*) and Immgen and Mouse RNAseq databases.

Differential expression analysis among the identified cell clusters was conducted utilizing the Find-Markers function, which implements a Wilcoxon Rank Sum test. Genes were deemed differentially expressed with an adjusted p-value <0.05 following correction for multiple testing via the Benjamini-Hochberg procedure.

## Bulk RNAseq

Naïve CD8-enriched lineage-depleted spleen cells from four male 4–6 week-old *Cd47*−/− and four male WT mice were prepared using CD8a+T Cell Isolation Kit, subjected to RNAseq analysis (*Nath et al., 2022*). Downstream analysis and visualization were performed within the NIH Integrated Analysis Platform (NIDAP) using R programs developed by a team of NCI bioinformaticians on the Foundry platform (Palantir Technologies). Briefly, RNA-seq FASTQ files were aligned to the reference genome (mm10) using STAR (*Dobin et al., 2013*) and raw counts data produced using RSEM (*Li and Dewey, 2011*). The gene counts matrix was imported into the NIDAP platform, where genes were filtered for low counts (<1 cpm) and normalized by quantile normalization using the limma package (*Ritchie et al., 2015*). Differentially expressed genes were calculated using limma-Voom (*Law et al., 2016*) and GSEA was performed using the fgsea package (*Korotkevich et al., 2019*). Preranked gene set enrichment analysis of the Hallmark collection was performed using the t-statistic as ranking variable. (*Figure 1—figure supplement 1*).

## Acknowledgements

We thank Dr. Michael Kelly at the Single Cell Analysis Facility for performing preliminary clustering and data analysis.

## Additional information

### Funding

| Funder | Grant reference number | Author |
|---|---|---|
| National Cancer Institute | ZIA SC009172 | David D Roberts |

The funders had no role in study design, data collection and interpretation, or the decision to submit the work for publication.

### Author contributions

Rajdeep Banerjee, Conceptualization, Formal analysis, Validation, Investigation, Visualization, Methodology, Writing – original draft, Writing – review and editing; Thomas J Meyer, Data curation, Software, Formal analysis, Visualization, Writing – review and editing; Margaret C Cam, Formal analysis, Project administration, Writing – review and editing; Sukhbir Kaur, Conceptualization, Formal analysis, Investigation, Methodology, Writing – original draft, Writing – review and editing; David D Roberts, Conceptualization, Formal analysis, Supervision, Funding acquisition, Writing – original draft, Project administration, Writing – review and editing

## Author ORCIDs

Rajdeep Banerjee (iD) http://orcid.org/0009-0007-0878-2034
Thomas J Meyer (iD) http://orcid.org/0000-0002-7185-5597
Margaret C Cam (iD) http://orcid.org/0000-0001-8190-9766
David D Roberts (iD) https://orcid.org/0000-0002-2481-2981

## Ethics

All animal experiments were carried out in strict accordance with the Recommendations for the Care and Use of Laboratory Animals of the National Institutes of Health under protocol LP-012 approved by the NCI Animal Care and Use Committee.

Reviewer #1 (Public Review): https://doi.org/10.7554/eLife.92679.3.sa1
Reviewer #3 (Public Review): https://doi.org/10.7554/eLife.92679.3.sa2
Author response https://doi.org/10.7554/eLife.92679.3.sa3

## Additional files

### Supplementary files

• MDAR checklist

### Data availability

Data supporting this publication have been deposited in NCBI's Gene Expression Omnibus and are accessible through GEO Series accession GSE239430. The code used to produce bioinformatics results can be found at (https://github.com/NIDAP-Community/SCWorkflow; copy archived at *NIDAP-Community, 2024* and https://github.com/NIDAP-Community/Regulation-of-Extramedullary-Erythropoiesis-by-CD47-and-THBS1; copy archived at *NIDAP-Community, 2023*). Primary flow cytometry data is available at https://zenodo.org/records/10904137. Additional data may be found in figure supplements available with the online version of this article.

The following datasets were generated:

| Author(s) | Year | Dataset title | Dataset URL | Database and Identifier |
|---|---|---|---|---|
| Banerjee R, Meyer TJ, Cam MC, Kaur S, Roberts DD | 2024 | Differential regulation by CD47 and thrombospondin-1 of extramedullary erythropoiesis in mouse spleen | https://www.ncbi.nlm.nih.gov/geo/query/acc.cgi?acc=GSE239430 | NCBI Gene Expression Omnibus, GSE239430 |
| Banerjee R, Meyer TJ, Cam MC, Kaur S, Roberts DD | 2024 | Differential regulation by CD47 and thrombospondin-1 of extramedullary erythropoiesis in mouse spleen | https://doi.org/10.5281/zenodo.10904137 | Zenodo, 10.5281/zenodo.10904137 |

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
