## [Editor Report · eLife assessment]

This study presents a **valuable** finding on the cell composition in mouse spleen depleted for the CD47 receptor and its signaling ligand Thrombospondin in hematopoietic differentiation. The supporting evidence is **convincing** with analytical improvements on the individual contributions of the signaling components and with functional studies. This work has implications for the role of CD47/Thbs1 in extramedullary erythropoiesis in mouse spleen and will be of interest to medical biologists working on cell signaling, transfusion medicine, and cell therapy.

---

## [Referee Report · Reviewer #1 (Public Review)]

Summary:

This study investigated the role of CD47 and TSP1 in extramedullary erythropoiesis by utilization of both global CD47-/- mice and TSP1-/- mice.

Strengths:

Flow cytometry combined with spleen bulk and single cell transcriptomics were employed. The authors found that stress-induced erythropoiesis markers were increased in CD47-/- spleen cells, particularly genes that are required for terminal erythroid differentiation. Moreover, CD47 dependent erythroid precursors population was identified by spleen scRNA sequencing. In contrast, the same cells were not detected in TSP1-/- spleen. These findings provide strong evidence to support the conclusion that differential role of CD47 and TSP1 in extramedullary erythropoiesis in mouse spleen. Furthermore, the relevance of the current finding to the prevalent side effect (anemia) of anti-CD47 mediated cancer therapy has been discussed in the Discussion section.

---

## [Referee Report · Reviewer #3 (Public Review)]

The authors used existing mouse models to compare the effects of ablating the CD47 receptor and its signaling ligand Thrombospondin. They analyze the cell composition of the spleens from CD47-KO and Thsp-KO using Flow Cytometry and single cell sequencing and focus mostly on early hematopoietic and erythroid populations. The data broadly shows that splenomegaly in the CD47-KO is largely due to an increase in committed erythroid progenitors, whereas the Thsp-KO shows a slight depletion of committed erythroid progenitors but is otherwise similar to WT in splenic cell composition. Thus, both their datasets supports the main conclusions of the study. One caveat of the single-cell dataset is that, insofar as the authors have explored and presented it, a clear picture of the mechanism driving extra medullary erythropoiesis in CD47-KO is lacking. This would be extremely valuable since one of the stated translational implications of this study is to assess and remedy the anemia caused by anti-CD47 therapy used in subtypes of AML. Nevertheless, this study provides novel insights into a putative role of Thsp-CD47 signaling in triggering definitive erythropoiesis in the mouse spleen in response to anemic stress and constitutes a good resource for researchers seeking to understand extramedullary erythropoiesis. This study also has generated data that will enable exploration of the possible adverse effects of using anti-CD47 therapies to treat AML.

---

## [Author Response]

The following is the authors’ response to the original reviews.

**Public Reviews:**

**Reviewer #1 (Public Review):**

Summary:This study investigated the role of CD47 and TSP1 in extramedullary erythropoiesis by utilization of both global CD47-/- mice and TSP1-/- mice.Strengths:Flow cytometry combined with spleen bulk and single-cell transcriptomics were employed. The authors found that stress-induced erythropoiesis markers were increased in CD47-/- spleen cells, particularly genes that are required for terminal erythroid differentiation. Moreover, CD47 dependent erythroid precursors population was identified by spleen scRNA sequencing. In contrast, the same cells were not detected in TSP1-/- spleen. These findings provide strong evidence to support the conclusion that the differential role of CD47 and TSP1 in extramedullary erythropoiesis in mouse spleen.Weaknesses:Methods and data analysis are appropriate. However, some clarifications are required. The discussion section needs to be expanded.(1) The sex of mice that were used in the study is unknown.(2) In the method of Single-cell RNA sequencing (page 10), it mentioned that single cell suspensions from mouse spleens were depleted of all mature hematopoietic cell lineages by passing through CD8a microbeads and CD8a+ T cell isolation Kit. As described, it is confusing what cell types are obtained for performing scRNAseq. More information is required for clarity.(3) The constitutive CD47 knockout mouse model is utilized in this study. The observed accumulation of erythroid precursors in the spleens of CD47-/- mice suggests a chronic effect of CD47 on spleen function. Can the current findings be extrapolated to acute scenarios involving CD47 knockdown or loss, as this may have more direct relevance to the potential side effects associated with an-CD47-mediated cancer therapy? Please expand on this topic in the discussion section.

(1) The missing mouse gender information is incorporated into the revised manuscript. For flow cytometry, two male and two female mice of each genotype were used. For single cell RNA sequencing, two female and one male mouse of each genotype were used. For the bulk RNA sequencing four male *cd47−/−* mice and four male wildtype mice were used.

(2) We apologize for the confusing presentation, which has been corrected. The bulk RNA sequencing analysis identified elevated expression of erythropoietic genes in CD8+ spleen cells from *cd47−/−* versus wildtype mice that were obtained using magnetic bead depletion of all other lineages. Therefore, we used the same Miltenyi negative selection kit as the first step to prepare the cells for single cell RNA sequencing. These untouched cells were then depleted of most mature CD8 T cells using a Miltenyi CD8a(Ly2) antibody positive selection kit. An important consideration underlying this approach was recognizing that the commercial magnetic bead depletion kits used for preparing specific immune cell types are optimized to give relatively pure populations of the intended immune cells using wildtype mice. Our previous experience studying NK cell development in the *cd47−/−* mice taught us that NK precursors, which are rare in wildtype mouse spleens, accumulate in *cd47−/−* spleens and were not removed by the antibody cocktail optimized for wildtype spleen cells (Nath et al Front Immunol 2018). The present data indicate that erythroid precursors behave similarly.

(3) The Discussion was edited as recommended. Anemia is a prevalent side effect of several CD47 therapeutic antibodies being developed for cancer therapy. This anemia would be expected to induce erythropoiesis in bone marrow and possibly at extramedullary sites. Human spleen cells are not accessible to directly evaluate extramedullary erythropoiesis in cancer patients, but analysis of circulating erythroid precursors or liquid biopsy methods could be useful to detect induction of extramedullary erythropoiesis by these therapeutics. We are currently investigating the ability of CD47 antibodies to directly induce erythropoiesis using a human in vitro model.

**Reviewer #2 (Public Review):**
Summary:The authors used existing mouse models to compare the effects of ablating the CD47 receptor and its signaling ligand Thrombospondin. The CD47-KO model used in this study was generated by Kim et al, 2018, where hemolytic anemia and splenomegaly was reported. This study analyzes the cell composition of the spleens from CD47-KO and Thsp-KO, focusing on early hematopoietic and erythroid populations. The data broadly shows that splenomegaly in the CD47-KO is largely due to an increase in committed erythroid progenitors as seen by Flow Cytometry and single-cell sequencing, whereas the Thsp-KO shows a slight depletion of committed erythroid progenitors but is otherwise similar to WT in splenic cell composition.Strengths:The techniques used are appropriate for the study and the data support the main conclusions of the study. This study provides novel insights into a putative role of Thsp-CD47 signaling in triggering definitive erythropoiesis in the mouse spleen in response to anemic stress and constitutes a good resource for researchers seeking to understand extramedullary erythropoiesis.Weaknesses:The Flow cytometry data alone supports the authors' main conclusion and single-cell sequencing confirms them but does not add further information, other than those already observed in the Flow data. The single-cell sequencing analysis and presentation could be improved by using alternate clustering methods as well as separating the data by genotype and displaying them in order for readers to fully grasp the nuanced differences in marker expression between the genotypes. Further, it is not clear from the authors' description of their results whether the increased splenic erythropoiesis is a direct consequence of CD47-KO or a response to the anemic stress in this mouse model. The enrichment of cKit+ Ter119+ Sca1- cells in CD47-KO indicates that these are likely stress erythroid progenitors. Another CD47-KO mouse model (Lindberg et al 1996) has no reported erythroid defects and was also not examined in this study.

(1) The reviewer asked, “whether the increased splenic erythropoiesis is a direct consequence of CD47-KO or a response to the anemic stress in this mouse model.” Our data supports both a direct role for CD47 and an indirect role resulting from the response to anemic stress. We cited our previous publications describing increased Sox2+ stem cells in spleens of *Cd47* and *Thbs1* knockout mice, but we neglected to emphasize another study where we found that bone marrow from *cd47−/−* mice subjected to the stress of ionizing radiation exhibited more colony forming units for erythroid (CFU-E) and burst-forming unit-erythroid (BFU-E) progenitors compared to bone marrow from irradiated wildtype mice (Maxhimer *Sci Transl Med* 2009). Taken together, our published data demonstrates that loss of CD47 results in an intrinsic protection of hematopoietic stem cells from genotoxic stress. This function of CD47 is thrombospondin-1-dependent and is consistent with the up-regulation of early erythroid precursors in the spleens of both knockout mice but cannot explain why the *Thbs1−/−* mice have fewer committed erythroid precursors than wildtype. We cited studies that documented increased red cell turnover in *cd47−/−* mice but less red cell turnover in *Thbs1−/−* mice compared to wildtype mice. Increased red cell clearance in *cd47−/−* mice is mediated by loss of the “don’t eat me” function of CD47 on red cells. In wildtype mice, clearance is augmented by thrombospondin-1 binding to the clustered CD47 on aging red cells (Wang, *Aging Cell* 2020). Thus, anemic stress in the mouse strains studied here decreases in the order *cd47−/−* > WT > *Thbs−/−*. This is consistent with the increased committed erythroid progenitors reported here in *cd47−/−* spleens and decreased committed progenitors in the *Thbs1−/−* spleens.

(2) Based on the reviewer’s question regarding alternative mechanisms and the publication of Yang et al 2022 identifying a role for CD47 in stress erythropoiesis though transfer of mitochondria to erythroblasts, we asked whether cd47-/- erythroid precursors would show decreased mRNA expression for mitochondrial chromosome genes (new Figure 4−figure supplement 3C). Some of these mRNAs were more abundant in cd47-/- and thbs1-/- erythroid cells, which is the opposite of what we expected based on Yang 2022 but consistent with our previous publications identifying thrombospondin-1 and CD47 as negative regulators of mitochondrial homeostasis in muscle cells and T cells.

(3) The *cd47−/−* mice used for the current study are the same strain as those reported by Lindberg et al in 1996, with additional backcrossing onto a C57BL/6 background.

**Recommendations For The Authors:**

**Reviewer #2 (Recommendations For The Authors):**
Suggestions for improved or additional experiments, data, or analyses.Significant efforts went into analyzing the type of erythroid progenitors by marker expression, but typical Flow cytometry strategies using Ter119 and CD44 combined with forward scatter can be used to stage the committed erythroid progenitors precisely.

We appreciate this suggestion to extend the flow data. However, the upcoming retirement of the PI required closing our breeding colony, and the mice are no longer available.

How can the difference between the erythroid phenotypes of the Lindberg et al 1996 CD47-KO (exon2 Neo knock-in) and Kim et al 2018 CD47-ko (exon1 26bp indel) be explained?

We are not convinced that the erythroid phenotypes of the Lindberg and Kim CD47-KO mice differ at the age used in our studies. Kim et al. focused on progressive hemolytic anemia and changes in T cells in spleen that emerge at 26 weeks age, whereas the mice used here were younger. The Lindberg and Kim mice have similar spleen enlargement at the age we used.

Another manuscript under review from our lab suggests that cis-regulation of an adjacent colinear gene could contribute to some phenotypes observed when perturbing the *Cd47* gene. The Lindberg mouse exhibits minimal perturbation of that adjacent gene, but we have no data regarding the Kim et al mouse. The reviewer’s question brought to our attention that we neglected to state in the Methods that the mice used here are the Lindberg mice, not the Kim mice. This omission is now corrected.

The authors used Lindberg mouse for 2018 study on NK cells and observed splenomegaly. Did they check for extramedullary erythropoiesis there?

Retrospective examination of the RNAseq data for the spleen cells enriched in NK precursors used in our 2018 publication (Nath, 2018) reveals significantly elevated expression for a majority of the extramedullary erythroid markers listed in Table 1, but they were generally less abundant than observed for the lineage-depleted spleen cells used in the present manuscript.

**Author response table 1. sa3table1:** 

Extramedullary	CD47KO/WT_FC	CD47KO/WT_pval	DEG rank
erythropoiesis Gene			
Ermap	3.202941868	0.035938856	2087
Gypa	23.09289299	0.000169485	163
Gata1	2.809105664	0.044897197	2273
Slc4a1	4.352812248	0.005732723	1641
Klf1	30.09543812	0.000442139	76
Trim10	10.8936042	0.003179996	672
Sptb	4.443091877	0.00032532	1607
Rhag	25.90869436	8.45 E-05	121

To clarify the stress erythropoiesis issue, it might be helpful to examine the sc-seq data for the expression of specific stress erythropoiesis markers in CD47-KO. Targets of BMP4 and Hedgehog signaling can also be examined. Further colony assays can help determine if stress BFU-Es are prevalent in the CD47-KO spleens and depleted in Thsp-KO

As noted in Table 1, twelve of the genes we studied are established markers of stress-induced extramedullary erythropoiesis, and most of these were included in the scRNA seq data presented. Our previous publication demonstrated that bone marrow from cd47−/− mice subjected to the stress of ionizing radiation exhibited more colony forming units for erythroid (CFU-E) and burst-forming unit-erythroid (BFU-E) progenitors compared to bone marrow from irradiated wildtype mice (Maxhimer Sci Transl Med 2009). We have not performed colony formation assays using spleen.

To address the reviewer’s question regarding BMP4 and hedgehog signaling we performed gene set enrichment analysis for known BMP4 and hedgehog signaling signatures. Using GSE26351_UNSTIM_VS_BMP_PATHWAY_STIM_HEMATOPOIETIC_PROGENITORS, cd47-/- cells in cluster 12 or their CD34+ orCD34- subsets did not show significant enrichment for BMP4 targets compared to WT. Thbs1-/- cells in clusters 12 and 14 showed marginally significant depletion of the BMP4 signature (p=0.04 and p=0.023, respectively). Using the KEGG_HEDGEHOG_SIGNALING_PATHWAY, we did not find any significant enrichment. However, only a few genes in this pathway were detectable in the scRNAseq data. These data suggest that the BMP4 signaling may be regulated by thrombospondin-1, but properly testing this hypothesis would require achieving greater sequencing depth combined with a cell isolation method that better enriches the early hematopoietic progenitors that are known to utilize the BMP4 pathway.

In the reclustering of erythroid progenitors in Figure 5, inclusion of Gata1 as a selection marker may help capture more of the early erythroid progenitors from the dataset and provide a more complete picture of the erythroid populations.

We thank the reviewer for suggesting inclusion of Gata1. We repeated the reclustering including Gata1 and found the selected cell count increased from 876 cells to 1007 cells. However, most of the increase was not in the erythroid cluster, which increased from 413 cells to 419 cells. Most of the increase represented Gata1+ T cells (548 cells including Gata1 versus 463 cells without). The revised manuscript presents genotype-dependent differential gene expression based on including Gata1 selection, but none of the specific conclusions were changed from the initial submission. The new Table 4 and Figure 7−figure supplement 1 enabled us to compare differential expression of erythropoietic genes obtained using supervised and unsupervised clustering and show that both methods yield comparable results.

Just out of curiosity, was there an attempt to make a CD47 Thsp double KO? . Is it viable?

Cd47 KO mice are somewhat difficult breeders, and several previous attempts to cross with other transgenics have produced viable homozygous offspring that could not be propagated.

Recommendations for improving the wring and presentation.Perhaps readers would find it more intriguing if the paper led with the single-cell sequencing showing enrichment of erythroid populations in CD47-KO, and later confirmed with Flow Cytometry (even if this was not necessarily the order in which the experiments were done).

We considered this suggestion but believe that some of the flow cytometry data is needed to understand why we focused on CD34+ and CD34- subsets and proliferation markers when analyzing the scRNAseq data

The single-cell sequencing data in Figure 3 might benefit from UMAP clustering as well. In addition, it would greatly help readers if the data points were separated by genotype and displayed after clustering. A similar analysis has been done in this paper: doi:10.1038/s41556-022-00898-9 by clustering different conditions together but displaying them separately by condition.

We initially explored tSNE and UMAP clustering and obtained similar results. We have added violin plots separated by genotype in Figure 4-figure supplement 2. We also included improved clusters separated by genotype in the revised Figure 3 panels C and D and for the reclustering in Figure 6D. UMAP plots provided better presentation for the reclustering (revised Figure 7). All data have been updated to the latest pipeline as noted in the Methods.

Minor corrections to the text and figures.Figure 4: Labels and plot legends are illegible in general, please relabel manually and if possible, redo plots with bigger font size and legends (relatively easy using ggplot2)

All figure panels were relabeled using larger fonts

Figure 5D: Individual plots are stacked randomly atop each other and in many cases, gene names are not visible. Please restack the layers and ensure that the gene names are visible

Panel D was made a separate figure with enlarged labels (now Figure 7).

Supp Fig 2: Layout can be organized a little better. Consider splitting into two figures for better organization

The figure was split as recommended. Now Figure 1-figure supplement 2 and Figure 2-figure supplement

1.

Abstract Line 10: "...mRNA expression of Kit, Ermap, and Tfrc, Induction of committed erythroid precursors is...". Replace comma after "Tfrc" with period

Done.

Discussion Page 9 Line 8: "...WT spleens, s. mRNAs for some markers of committed erythroid cells including Nr3c1 mRNA...". Remove ", s" after spleens.

Done.